# Structure-Aware Riemannian Flow Matching for Registration and Fusion of Hyperspectral and Multispectral Images

**Quan Zhang** [1]  **Jun Li** [2]  **Weilong Zhu** [1]  **Mingyang Li** [2]  **Qinmu Shen** [2]  **Yuanxi Peng** [1]

## Abstract

Precise alignment is a prerequisite for hyperspectral and multispectral image fusion, yet existing methods struggle with complex non-rigid deformations. Existing techniques either suffer from inter-task error accumulation by treating registration and fusion as disjoint processes or neglect the geometric nature of distortions by relying on isotropic Euclidean metrics. We propose Structure-Aware Riemannian Flow Matching (SA-RFM), a geometry-informed framework for joint registration and fusion of hyperspectral and multispectral images. SA-RFM reformulates registration as dynamic optimal transport on a structure-induced Riemannian manifold, where anisotropic costs are derived from MSI structural cues. To circumvent the complexity of explicit OT solvers, we incorporate this geometry into a conditional flow matching framework via a local cost approximation. This formulation is realized through two synergistic mechanisms: a Riemannian Flow Matching objective that enforces structure-aware error measurement, and an optimal transport direction regularization that aligns the velocity field with the induced metric, thereby resolving the fundamental mismatch between anisotropic costs and conventional Euclidean supervision. Extensive experiments on four datasets demonstrate the superiority of our method. Code is available at: https://github.com/ZhangQuan-hub.

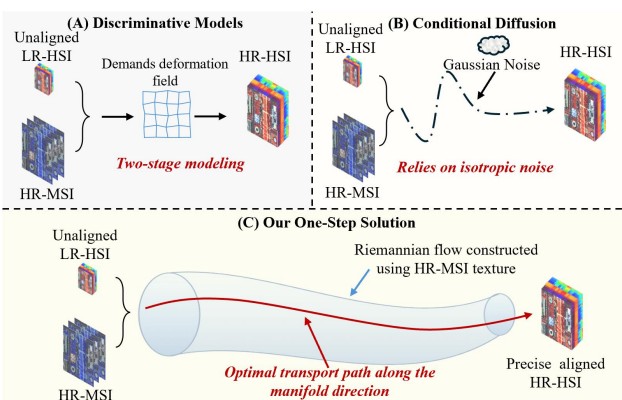

*Figure 1.* Conceptual comparison of different deep-learning-based registration and fusion methods. (A) Discriminative methods: These require explicit modeling of the deformation field, where the two-stage optimization is prone to local optima. (B) Conditional Diffusion: These methods rely on isotropic Gaussian noise, which lacks physical priors. (C) Ours: By integrating texture-informed anisotropic Riemannian manifolds with optimal transport optimization, our approach achieves unified one-step modeling.

## 1. Introduction

Hyperspectral images (HSIs), which capture scenes across hundreds of narrow and contiguous spectral bands, provide rich spectral information that is invaluable for a wide range of applications, including precision agriculture (Sahoo et al., 2015; Lu et al., 2020; Ang & Seng, 2021), environmental monitoring (Stuart et al., 2019; Peyghambari & Zhang, 2021), food safety (Feng & Sun, 2012; Pu et al., 2023), mineral exploration (Bishop et al., 2011; Pour et al., 2021), and remote sensing (Zhu et al., 2023; Dian et al., 2026; Liu et al., 2025). However, due to fundamental hardware constraints, hyperspectral imaging systems inevitably face a trade-off between spatial and spectral resolution. As a result, HSIs typically exhibit low spatial resolution, which severely limits their practical usability.

To alleviate this limitation, multispectral and hyperspectral image fusion (MHIF) has emerged as a key technique. By combining the rich spatial details of a high-resolution multispectral image (HR-MSI) with the abundant spectral information of a low-resolution HSI (LR-HSI), MHIF aims to reconstruct a high-resolution HSI (HR-HSI) (Luo et al.,

[1]College of Computer Science and Technology, National University of Defense Technology, Changsha, China [2]College of Electronic Science and Technology, National University of Defense Technology, Changsha, China. Correspondence to: Jun Li <lijun14@nudt.edu.cn>.

*Proceedings of the 43$^{rd}$ International Conference on Machine Learning*, Seoul, South Korea. PMLR 306, 2026. Copyright 2026 by the author(s).

2025; Ma et al., 2024). Crucially, the success of MHIF critically depends on accurate pixel-level alignment between the MSI and HSI. In real-world scenarios, however, such image pairs are often affected by complex non-rigid misalignments caused by platform vibrations, sensor parallax, and atmospheric turbulence. Consequently, robust cross-modality registration remains a fundamental yet highly challenging prerequisite for reliable hyperspectral image fusion.

Traditional registration methods (Lowe, 2004; Zhou et al., 2020) often fail under severe modality gaps and have largely been superseded by deep learning approaches. Existing learning-based methods can be broadly divided into two paradigms. The first paradigm consists of discriminative regression models based on convolutional neural networks (Dong et al., 2021; Zheng et al., 2021; Qu et al., 2025a;c), transformers (Dong et al., 2024) or Mamba-based methods (Li et al., 2025). As shown in Figure 1, these methods explicitly reconstruct pixel-level deformation fields for registration, upon which image fusion is performed subsequently. However, such two-stage optimization is prone to task-switching errors, which can cause the model to fall into local minima and render it unable to handle complex distortion types. The second paradigm leverages generative models (Ho et al., 2020; Song et al., 2020; Rombach et al., 2022), which offer stronger image prior modeling capabilities. Diffusion models are one of the more popular methods among them (Qu et al., 2025b; Diao et al., 2025). However, standard diffusion-based formulations rely on isotropic Gaussian noise priors. When applied to registration, this assumption introduces a fundamental geometric mismatch: structured, directional misalignments are treated as isotropic noise. Consequently, the denoising trajectory is forced to traverse a noise-dominated space that ignores image topology, often resulting in spectral mixing and blurred structural boundaries.

We argue that image registration should instead be viewed as a transport problem governed by geometry-aware costs. Specifically, transporting pixels across strong structural boundaries should incur significantly higher costs than motion along coherent structures. This motivates a formulation based on Optimal Transport (OT) with anisotropic, data-dependent costs. However, explicitly solving OT or computing geodesics on high-dimensional image manifolds is computationally intractable.

To address these challenges, we propose Structure-Aware Riemannian Flow Matching (SA-RFM), a geometry-informed learning framework for the registration and fusion of HSIs and MSIs. SA-RFM models image registration as a dynamic optimal transport process on a structure-induced Riemannian manifold, where anisotropic transport costs are derived from MSI structural cues. Rather than explicitly solving optimal transport or computing geodesics, we

incorporate the induced geometry into a conditional flow matching framework via a locally approximated transport cost. SA-RFM introduces two complementary geometric mechanisms. First, a Riemannian Flow Matching objective enforces structure awareness at the error-measurement level by penalizing velocity deviations along high-cost directions. Second, we identify a fundamental mismatch between anisotropic transport costs and conventional Euclidean displacement supervision, and resolve it by introducing a structure-aware optimal transport direction regularization. This regularization explicitly aligns the learned velocity field with minimum-cost transport directions under the induced Riemannian metric, leading to structure-consistent deformations without sacrificing computational efficiency.

**Main Novelty and Contributions.** **1)** We formulate the unaligned HSI and MSI fusion task as a geometry-aware dynamic optimal transport problem, and demonstrate how optimal transport principles can be integrated into learning-based registration without explicitly solving OT or computing geodesics. **2)** We propose a structure-aware Riemannian flow matching framework that introduces two complementary geometric mechanisms: a Riemannian flow matching objective that enforces structure-aware error measurement, and an optimal transport direction regularization that resolves the fundamental mismatch between anisotropic transport costs and conventional Euclidean displacement supervision. **3)** Extensive experiments on multiple hyperspectral benchmarks show that SA-RFM consistently achieves superior alignment accuracy and spectral fidelity, while maintaining high computational efficiency compared to existing discriminative and generative registration and fusion approaches.

**Conflict of Interest Disclosure.** The authors declare no financial conflicts of interest related to this work.

## 2. Preliminaries

We provide a brief review of Optimal Transport and Conditional Flow Matching. Related work is discussed in Appendix C.

### 2.1. Optimal Transport

Optimal Transport (OT) provides a principled framework for comparing and transforming probability measures by minimizing transport costs. In the context of image registration, Monge OT is theoretically superior to stochastic relaxations (such as the Kantorovich problem) because physical registration necessitates a point-to-point correspondence (i.e., deterministic deformation) rather than mass splitting or diffusion. We focus on the **Monge problem**, which seeks a deterministic mapping $T : \Omega \rightarrow \Omega$ (where $\Omega \subset \mathbb{R}^2$ denotes the spatial domain and $u \in \Omega$ is the spatial coor-

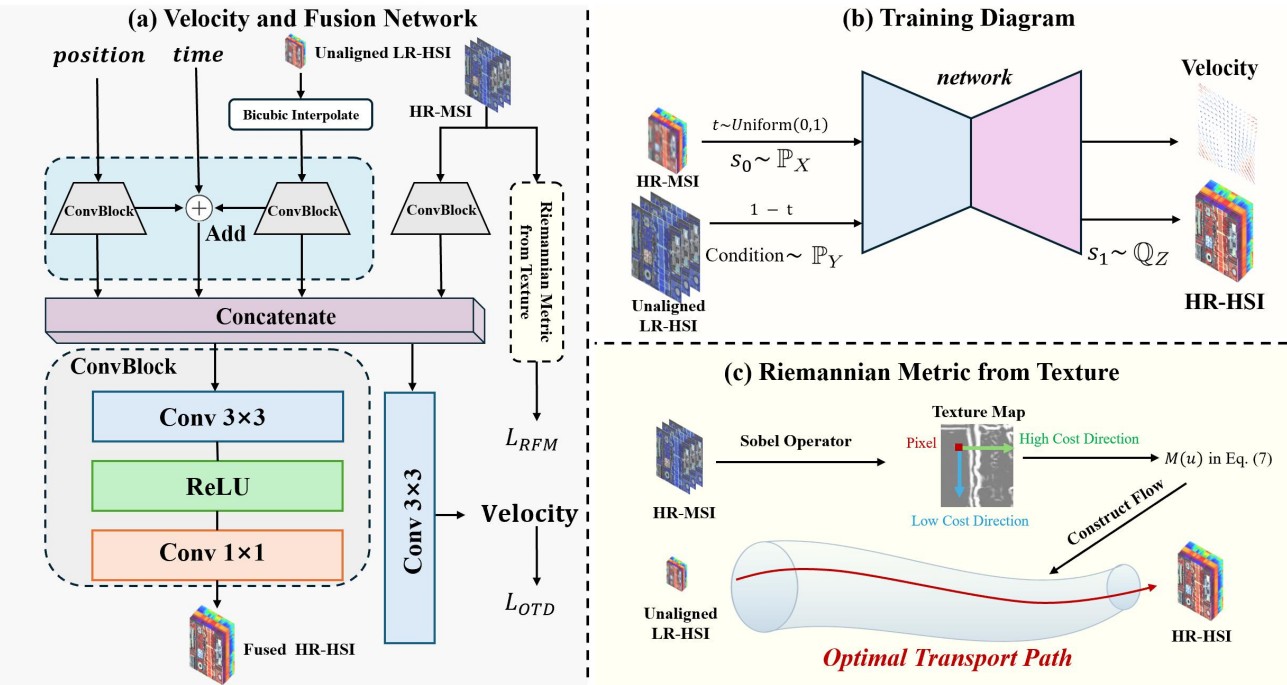

*Figure 2.* The proposed SA-RFM framework. (a) Architecture of the fusion modules and velocity network, where both the encoders and decoder are constructed using ConvBlocks. (b) The training scheme involving data sampling at various time steps. (c) Illustration of the proposed Riemannian metric formulated in Eq. (7), which assigns higher transport costs across texture directions and lower costs along them.

dinate vector) that pushes the source distribution $\mathbb{P}$ toward the target distribution $\mathbb{Q}$. This process is described by the push-forward operator $T_\#$: $T_\#\mathbb{P} = \mathbb{Q}$, representing the evolution of the measure under the mapping. The optimization objective is to minimize the total transport cost:

$$\mathcal{C}(T) = \int_\Omega c(u, T(u))\, d\mathbb{P}(u), \tag{1}$$

where $c(u, T(u))$ encodes the geometric cost of moving mass from $u$ to $T(u)$, typically taken as the squared Euclidean distance in traditional frameworks.

The **Benamou-Brenier formula** establishes an equivalent dynamic perspective for the Monge problem. It represents the transport process through a time-varying velocity field $v_t$, governed by the ordinary differential equation (ODE) $\frac{ds_t}{dt} = v_t(s_t)$, where $s_t$ denotes the particle trajectory and $t \in [0, 1]$ is the time parameter. In this dynamic view, the Monge problem is equivalent to minimizing the average kinetic energy of the flow over the entire time interval:

$$\inf_{v_t} \int_0^1 \int_\Omega \frac{1}{2} \|v_t(u)\|_2^2\, \rho_t(u)\, du\, dt, \tag{2}$$

subject to the continuity equation $\partial_t \rho_t + \nabla \cdot (\rho_t v_t) = 0$, where $\rho_t(u)$ denotes the probability density function of the transported measure at time $t$ and position $u$. This dynamic perspective bridges OT with generative models, enabling

efficient optimization on large-scale image data via flow matching schemes.

## 2.2. Conditional Flow Matching

**Conditional Flow Matching (CFM)** is a method for learning the velocity field $v_t$ that generates a probability path between the starting distribution $\mathbb{P}$ and the target distribution $\mathbb{Q}$. While directly learning the total vector field of marginal probability paths is extremely difficult, we can learn a conditional vector field conditioned on specific sample pairs $(s_0, s_1)$. For a linear interpolation path $\psi_t(u|s_0, s_1) = (1 - t)s_0 + ts_1$, the corresponding conditional vector field (i.e., conditional velocity) is constant: $u_t(s_t|s_0, s_1) = s_1 - s_0$. By integrating over all sample pairs in the training set, the CFM objective function is defined as:

$$\mathcal{L}_{\text{CFM}}(\theta) = \mathbb{E}_{t, q(s_0, s_1), s_t} \left[ \|v_\theta(s_t, t) - (s_1 - s_0)\|^2 \right], \tag{3}$$

where $q(s_0, s_1)$ is the joint distribution of the source and target samples, and $v_\theta(s_t, t)$ represents the velocity field approximated by a neural network with parameters $\theta$. CFM converts the complex optimization into a regression task for a network representing the time-dependent velocity.

# 3. Methodology

We propose Structure-Aware Riemannian Flow Matching (SA-RFM), a geometry-informed learning framework that explicitly preserves fine image structures during flow matching by constructing a structure-induced Riemannian geometry. Our framework is shown in Figure 2.

## 3.1. Problem Formulation

**Geometric Modeling of Unregistered Image Fusion.** Consider an unregistered low-resolution hyperspectral image (LR-HSI) $\mathbf{X}_\downarrow \in \mathbb{R}^{h \times w \times C}$ and a high-resolution multispectral image (HR-MSI) $\mathbf{Y} \in \mathbb{R}^{H \times W \times c}$. Our goal is to reconstruct an aligned high-resolution hyperspectral image $\mathbf{Z} \in \mathbb{R}^{H \times W \times C}$ that preserves both spectral information and spatial structures.

From a transport perspective, we treat the upsampled LR-HSI $\mathbf{X} \in \mathbb{R}^{H \times W \times C}$ as the source distribution $s_0 \sim \mathbb{P}_X$, the desired aligned HR-HSI as the target distribution $s_1 \sim \mathbb{Q}_Z$, and the MSI as a reference distribution $y \sim \mathbb{P}_Y$ providing reliable structural priors. The registration–fusion operator is defined as:

$$\mathbb{Q}_Z = \mathcal{F}(T_\# \mathbb{P}_X, \mathbb{P}_Y), \qquad (4)$$

where $T : \Omega \to \Omega$ denotes the spatial transport map and $\mathcal{F}$ represents the fusion operator. In this formulation, $T$ corrects geometric distortions in $\mathbb{P}_X$ via the push-forward operator $T_\#$, while $\mathcal{F}$ subsequently fuses high-frequency spatial details from the HR-MSI.

## 3.2. Structure-Aware Riemannian Flow Matching

Due to the high dimensionality and severe ill-posedness of image registration, we adopt the *minimum-cost principle* as a physical constraint. Registration is interpreted as transporting spectral vectors in the spatial domain, seeking a smooth and invertible map $T^*$ that minimizes a cumulative transport cost:

$$T^* = \arg\min_T \int_\Omega c(u, T(u)) \, \mathrm{d}u. \qquad (5)$$

The design of the cost function $c$ is central to preserving the semantic integrity of the scene. We observe that traditional isotropic Euclidean transport costs fail to perceive complex scene structures. In unregistered scenarios, if pixels cross significant physical boundaries, such as road edges or building contours, during the transport process, severe spectral mixing and ghosting artifacts may occur. Therefore, an ideal transport process should be *anisotropic*: movements across structural boundaries should incur high costs, while motions along structural directions should be encouraged. To capture this spatial heterogeneity, we construct a structure tensor

from the MSI as

$$S_{\mathbf{Y}}(u) = \sum_{k=1}^{c} \left( \nabla \mathbf{Y}^{(k)}(u) \nabla \mathbf{Y}^{(k)}(u)^\top \right) * G_{k,\sigma}, \quad (6)$$

where $\nabla$ denotes the gradient, which is extracted using the Sobel operator. The ablation study evaluating various gradient computation methods is illustrated in Table 6. $\mathbf{Y}^{(k)}$ denotes the $k$-th MSI band and $G_{k,\sigma}$ is a Gaussian smoothing kernel with kernel size $k$ and standard deviation $\sigma$. The ablation studies on different construction methods of $S_{\mathbf{Y}}(u)$ and the parameters of the Gaussian kernel are shown in Table 6 and Figure 5.

Based on this tensor, we define a position-dependent Riemannian metric:

$$M(u) = I + \beta S_{\mathbf{Y}}(u), \qquad (7)$$

where $I$ denotes the $2 \times 2$ identity matrix representing the isotropic baseline and $\beta > 0$ is a scalar parameter controlling the sensitivity to scene structures. The exact formulation of the structure tensor is provided in Appendix B.4.

Rather than numerically solving the static assignment problem, we adopt a dynamic formulation to characterize the transformation flow. The transport map is represented by a continuous deformation $s_t$ governed by a time-dependent velocity field $v_t$:

$$\frac{\mathrm{d}s_t}{\mathrm{d}t} = v_t(s_t), \quad s_0 = u, \quad s_1 = T(u). \qquad (8)$$

Under the geometric view induced by $M(u)$, the optimal transport task is equivalent to finding a velocity field that minimizes the accumulated *structure-aware kinetic energy*:

$$\mathcal{J}(v_t) = \int_0^1 \int_\Omega v_t(u)^\top M(u) v_t(u) \rho_t(u) \mathrm{d}u \mathrm{d}t. \quad (9)$$

This formulation ensures that the registration path is not only smooth and invertible but also respects the underlying semantic boundaries of the HR-MSI. Derivation of Eq. (9) is provided in Appendix B.1.

Under this metric, the transport cost between two spatial locations: source location $u$ at LR-HSI and destination location $w$ at HR-HSI is naturally defined by the Riemannian energy:

$$c(u, w) = \inf_{\substack{\gamma : [0,1] \to \Omega \\ \gamma(0) = u, \ \gamma(1) = w}} \int_0^1 \dot{\gamma}(t)^\top M(\gamma(t)) \dot{\gamma}(t) \, dt, \quad (10)$$

where $\gamma(t)$ denotes a continuous path connecting $u$ and $w$, and $\dot{\gamma}(t) = \frac{d\gamma}{dt} \in T_{\gamma(t)}\Omega$ denotes the tangent velocity of the path at time $t$.

---

**Algorithm 1** The Training Process of Our Structure-Aware Riemannian Flow Matching (SA-RFM)

---

**Require:** Dataset $\mathcal{D}$ of tuples $(\mathbf{X}_\downarrow, \mathbf{Y}, \mathbf{Z})$ (LR-HSI, HR-MSI, HR-HSI); epochs $E$; metric params $(\beta, k, \sigma)$; weight $\lambda_{\text{OTD}}$.
**Ensure:** Trained velocity network $f_\theta$.
1: Initialize $\theta$ and optimizer.
2: **for** $e = 1, \ldots, E$ **do**
3:    Sample mini-batch $(\mathbf{X}_\downarrow, \mathbf{Z}, \mathbf{Y}) \sim \mathcal{D}$.
4:    $x_0 \leftarrow \text{Bicubic}(\mathbf{X}_\downarrow), \quad x_1 \leftarrow \mathbf{Z}$.
5:    Sample $t \sim \mathcal{U}(0, 1), \quad x_t \leftarrow (1 - t)x_0 + tx_1$, $u \leftarrow x_1 - x_0$.
6:    Compute structure tensor $S_\mathbf{Y}$ from $\mathbf{Y}$ (Sobel + Gaussian $G_{k,\sigma}$); metric $M \leftarrow I + \beta S_\mathbf{Y}$.
7:    Predict velocity $v_\theta \leftarrow f_\theta(x_t, t, \mathbf{X}_\downarrow, \mathbf{Y})$.
8:    $\mathcal{L}_{\text{RFM}} \leftarrow \mathbb{E}\big[(v_\theta - u)^\top M(v_\theta - u)\big]$.
9:    $v^\dagger \leftarrow \arg\min_v v^\top M v \quad \text{s.t.} \quad \langle v, u \rangle = \|u\|_2$; $\mathcal{L}_{\text{OTD}} \leftarrow \mathbb{E}\big[\|v_\theta - v^\dagger\|_2^2\big]$.
10:    Update $\theta$ by minimizing $\mathcal{L} = \mathcal{L}_{\text{RFM}} + \lambda_{\text{OTD}}\mathcal{L}_{\text{OTD}}$.
11: **end for**

---

For small displacements $\Delta u = w - u$, we approximate the above energy by freezing the metric at $u$ and assuming a straight-line path, leading to the following local quadratic form:

$$c(u, u + \Delta u) \approx \Delta u^\top M(u) \Delta u. \tag{11}$$

This approximation preserves directional selectivity while remaining computationally efficient. More ablation results on different metric constructions are reported in Table 7.

To learn the deformation flow, we adopt Conditional Flow Matching (CFM). In one-step CFM, the conventional supervision uses the Euclidean displacement: $v^*(u) = s_1(u) - s_0(u)$, here $v^*(u)$ represents the target velocity vector at location $u$. To incorporate structural constraints, we introduce the Riemannian Flow Matching (RFM) loss:

$$\mathcal{L}_{\text{RFM}} = \mathbb{E}_{t,s_0,s_1}\Big[ \\ \big(v_\theta(s_t, t) - v^*(u)\big)^\top M(s_t)\big(v_\theta(s_t, t) - v^*(u)\big)\Big]. \tag{12}$$

This loss penalizes errors along high-cost directions and encourages the learned velocity field to respect structures.

### 3.3. Optimal Transport Direction Regularization

While $\mathcal{L}_{\text{RFM}}$ introduces structure awareness at the error-measurement level, its supervision target remains the Euclidean displacement $v^*(u)$. In an anisotropic space induced by $M(u)$, however, the Euclidean direction is not necessarily the minimum-cost transport direction.

To address this issue, we first define the Euclidean displacement field as $d(u) = s_1(u) - s_0(u)$, representing the direct

vector difference between target and source at location $u$. We then define a *structure-aware optimal one-step transport direction* $v^\dagger(u)$ as the solution to:

$$v^\dagger(u) = \arg\min_v v^\top M(u)v, \\ \text{s.t.} \quad \langle v, d(u) \rangle = \|d(u)\|_2. \tag{13}$$

This formulation preserves the global direction toward the target (fixed by the inner product constraint) while selecting the minimum-cost direction under the Riemannian metric.

The above problem admits a closed-form solution:

$$v^\dagger(u) = \frac{M(u)^{-1}d(u)}{d(u)^\top M(u)^{-1}d(u)}\|d(u)\|_2. \tag{14}$$

The derivation from Eq. (13) to Eq. (14) is provided in Appendix B.2. Geometrically, $v^\dagger$ can be seen as a metric-correction of the Euclidean displacement that favors low-cost structural directions.

We incorporate the optimal transport direction as an additional regularization term:

$$\mathcal{L}_{\text{OTD}} = \mathbb{E}_u \big[\|v_\theta(u, t) - v^\dagger(u)\|_2^2\big], \tag{15}$$

where $v_\theta(u, t)$ denotes the neural network prediction at location $u$ and arbitrary time $t$ sampled uniformly in $[0, 1]$ during training.

The overall training objective is:

$$\mathcal{L} = \mathcal{L}_{\text{RFM}} + \lambda_{\text{OTD}}\mathcal{L}_{\text{OTD}}, \tag{16}$$

where $\lambda_{\text{OTD}}$ is the regularization coefficient balancing Euclidean consistency and structure-aware directional optimality (set to $1.0$ in our experiments). Sensitivity analysis is shown in Figure 4. Our implementation is summarized in Algorithm 1.

## 4. Experiments

### 4.1. Dataset Preparation

**Datasets.** We conducted extensive experiments on four hyperspectral datasets with varying spectral bands: Botswana, Pavia University, Chikusei, and Xiongan. The Botswana dataset contains $1476 \times 256$ pixels with a 30-m spatial resolution, covering 145 bands in the $0.4$–$2.5\,\mu\text{m}$ range. The Pavia University dataset consists of $610 \times 340$ pixels with a 1.3-m spatial resolution, comprising 103 bands ranging from $0.43$ to $0.86\,\mu\text{m}$. The Chikusei dataset features $2517 \times 2335$ pixels with a 2.5-m spatial resolution, covering 128 bands across $0.36$–$1.02\,\mu\text{m}$. The Xiongan dataset includes $3750 \times 1580$ pixels with a 0.5-m spatial resolution, spanning 256 spectral bands from $0.39$ to $1.00\,\mu\text{m}$.

*Table 1.* Quantitative comparison on the **Pavia University** dataset under Elastic, Homography, and Turbulence distortions. The best metrics are highlighted in **Bold** and the second-best are Underlined. The gray column highlights our proposed method.

| DISTORTION | METRIC | Integrated (TGRS'20) | MoGDCN (TIP'21) | NonRegSR (TGRS'22) | M2DTN (AAAI'24) | MoEPNP (TNNLS'24) | IR&ArF (TIP'25) | PSRFDif (TGRS'25) | RFDif (TGRS'25) | PRFCo (TGRS'25) | Ours |
|---|---|---|---|---|---|---|---|---|---|---|---|
| Elastic ($\alpha$=600, $\sigma$=15) | PSNR↑ | 40.66 | 36.22 | 39.43 | 37.19 | 40.13 | 39.43 | 41.49 | 37.77 | 42.33 | **42.69** |
| | SSIM↑ | 0.981 | 0.961 | 0.976 | 0.968 | 0.979 | 0.978 | 0.982 | 0.971 | 0.983 | **0.987** |
| | SAM↓ | 2.163 | 3.610 | 2.629 | 3.219 | 2.376 | 2.697 | 2.015 | 2.897 | 1.759 | **1.735** |
| | ERGAS↓ | 1.167 | 1.941 | 1.339 | 1.736 | 1.252 | 1.332 | 1.067 | 1.607 | 0.970 | **0.931** |
| | RMSE↓ | 2.363 | 3.942 | 2.724 | 3.522 | 2.512 | 2.724 | 2.148 | 3.298 | 1.951 | **1.872** |
| Homography (MPO=10%) | PSNR↑ | 40.85 | 38.39 | 39.16 | 35.59 | 40.96 | 36.73 | 41.03 | 38.88 | 42.33 | **42.57** |
| | SSIM↑ | 0.981 | 0.975 | 0.975 | 0.957 | 0.981 | 0.968 | 0.982 | 0.974 | 0.983 | **0.987** |
| | SAM↓ | 2.165 | 2.873 | 2.676 | 3.833 | 2.115 | 3.522 | 2.078 | 2.684 | 1.758 | **1.758** |
| | ERGAS↓ | 1.142 | 1.508 | 1.381 | 2.073 | 1.134 | 1.837 | 1.118 | 1.419 | 0.972 | **0.941** |
| | RMSE↓ | 2.312 | 3.071 | 2.811 | 4.239 | 2.283 | 3.718 | 2.266 | 2.900 | 1.951 | **1.898** |
| Turbulence ($D/r_0$=2.0) | PSNR↑ | 40.99 | 38.39 | 39.99 | 39.61 | 41.18 | 39.86 | 41.41 | 39.03 | 42.62 | **43.10** |
| | SSIM↑ | 0.981 | 0.975 | 0.977 | 0.977 | 0.981 | 0.980 | 0.982 | 0.977 | 0.984 | **0.988** |
| | SAM↓ | 2.137 | 2.873 | 2.459 | 2.543 | 2.078 | 2.468 | 2.051 | 2.622 | 1.722 | **1.658** |
| | ERGAS↓ | 1.120 | 1.508 | 1.256 | 1.316 | 1.108 | 1.267 | 1.073 | 1.394 | 0.939 | **0.886** |
| | RMSE↓ | 2.275 | 3.071 | 2.554 | 2.666 | 2.225 | 2.593 | 2.169 | 2.851 | 1.886 | **1.785** |
| Params (M) | | – | 7.918 | 1.900 | 17.64 | 45.42 | 23.20 | 66.24 | 14.72 | 8.367 | **1.078** |
| FLOPs (G) | | – | 567.6 | 62.32 | 169.9 | 905.0 | 1260 | 34.67 | 113.6 | 181.5 | **21.29** |

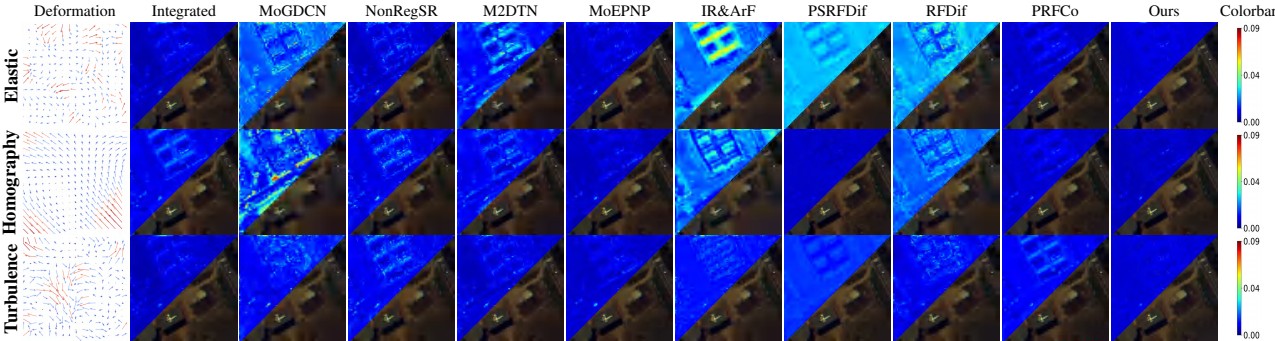

*Figure 3.* Visual comparison on the Pavia University dataset. The first column displays the deformation fields, followed by the recovery results of various methods and the error Colorbar. The top-left corner displays the difference map, while the bottom-right corner shows the RGB (bands 67-29-1) image.

**Dataset Splitting.** Following (Zhang et al., 2020), we designated the central $128 \times 128$ region of each dataset as the test set. During training, this test region was masked to zero, and training samples of size $128 \times 128$ were randomly cropped from the remaining areas. Training data were generated following Wald's protocol (Wald et al., 1997). The LR-HSI was created by applying a $5 \times 5$ Gaussian blur (standard deviation $\sigma = 2$) followed by $4\times$ bilinear downsampling. The HR-MSI was generated using the spectral response function of the Gaofen-1 satellite (Chen et al., 2022).

**Distortion Simulation.** To evaluate robustness, we applied different degrees of distortion to the LR-HSI, including: (i) *Elastic distortion* with parameters $(\alpha, \sigma) = (600, 15)$; (ii) *Homography distortion* with Maximum Perturbation Offset (MPO) rates of 10%; and (iii) *Turbulence distortion* (Mao et al., 2021) with the ratio of aperture diameter to the Fried parameter, $D/r_0 = 2.0$. Visualizations of specific

deformation fields are shown in Figure 3. Please refer to the Appendix for additional experimental results.

### 4.2. Benchmark and Evaluation Metrics

**Benchmark.** We compared our method with publicly available state-of-the-art approaches, categorizing them into: (1) Traditional method: Integrated (Zhou et al., 2020); (2) CNN-based methods: MoGDCN (Dong et al., 2021), Non-RegSR (Zheng et al., 2021), and MoEPNP (Qu et al., 2025a); IR&ArF (Qu et al., 2025c); (3) Transformer-based method: M2DTN (Dong et al., 2024), (4) Mamba-based method: PRFCo (Li et al., 2025); and (5) Diffusion-based methods: PSRFDif (Qu et al., 2025b) and RFDif (Diao et al., 2025).

**Metrics.** Performance is evaluated using five standard quantitative metrics: Peak Signal-to-Noise Ratio (PSNR) (Hore & Ziou, 2010), Structural Similarity Index (SSIM) (Wang

*Table 2.* Quantitative comparison on the **Botswana** dataset under Elastic, Homography, and Turbulence distortions. **Bold** indicates best, Underline indicates second-best. The gray column highlights our proposed method.

| DISTORTION | METRIC | Integrated (TGRS'20) | MoGDCN (TIP'21) | NonRegSR (TGRS'22) | M2DTN (AAAI'24) | MoEPNP (TNNLS'24) | IR&ArF (TIP'25) | PSRFDif (TGRS'25) | RFDif (TGRS'25) | PRFCo (TGRS'25) | Ours |
|---|---|---|---|---|---|---|---|---|---|---|---|
| Elastic ($\alpha$=600, $\sigma$=15) | PSNR↑ | 41.17 | 42.71 | 40.16 | 38.29 | 43.32 | 45.70 | 39.57 | 45.15 | 44.23 | **46.45** |
| | SSIM↑ | 0.955 | 0.987 | 0.980 | 0.970 | 0.989 | 0.993 | 0.978 | 0.992 | 0.988 | **0.995** |
| | SAM↓ | 5.985 | 3.651 | 4.232 | 5.252 | 3.580 | 2.573 | 5.381 | 2.797 | 3.657 | **2.411** |
| | ERGAS↓ | 1.360 | 1.142 | 1.530 | 1.872 | 1.066 | 0.806 | 1.650 | 0.867 | 0.966 | **0.743** |
| | RMSE↓ | 2.229 | 1.867 | 2.503 | 3.105 | 1.739 | 1.322 | 2.679 | 1.410 | 1.567 | **1.214** |
| Homography (MPO=10%) | PSNR↑ | 40.90 | 40.00 | 40.81 | 39.31 | 43.85 | 40.74 | 39.01 | 41.46 | 43.28 | **43.93** |
| | SSIM↑ | 0.970 | 0.976 | 0.982 | 0.978 | **0.990** | 0.960 | 0.974 | 0.967 | 0.986 | 0.977 |
| | SAM↓ | 4.629 | 4.813 | 4.121 | 4.810 | 3.376 | 4.342 | 5.999 | 4.014 | 3.897 | **3.235** |
| | ERGAS↓ | 1.408 | 1.550 | 1.420 | 1.713 | 1.003 | 1.457 | 1.737 | 1.319 | 1.061 | **0.983** |
| | RMSE↓ | 2.299 | 2.549 | 2.323 | 2.761 | 1.637 | 2.342 | 2.859 | 2.157 | 1.748 | **1.622** |
| Turbulence ($D/r_0$=2.0) | PSNR↑ | 41.42 | 43.51 | 41.48 | 41.84 | 43.27 | 45.92 | 40.28 | 45.34 | 44.14 | **46.69** |
| | SSIM↑ | 0.961 | 0.988 | 0.984 | 0.984 | 0.989 | 0.993 | 0.979 | 0.993 | 0.988 | **0.995** |
| | SAM↓ | 4.965 | 3.374 | 3.837 | 4.167 | 3.536 | 2.519 | 5.504 | 2.647 | 3.660 | **2.38** |
| | ERGAS↓ | 1.330 | 1.039 | 1.318 | 1.268 | 1.067 | 0.790 | 1.522 | 0.861 | 0.976 | **0.72** |
| | RMSE↓ | 2.165 | 1.702 | 2.150 | 2.063 | 1.750 | 1.290 | 2.469 | 1.379 | 1.582 | **1.18** |
| Params (M) ↓ | | – | 9.015 | 1.907 | 27.79 | 45.97 | 29.30 | 94.98 | 16.27 | 16.49 | **1.203** |
| FLOPs (G) ↓ | | – | 588.4 | 62.53 | 323.9 | 1038 | 1444 | 65.46 | 134.8 | 357.8 | **24.02** |

et al., 2004), Spectral Angle Mapper (SAM) (Yuhas et al., 1992), Dimensionless Global Error in Synthesis (ERGAS) (Wald, 2000), and Root Mean Square Error (RMSE).

### 4.3. Implementation Details

We trained the network for 400 iterations, utilizing 24 randomly cropped regions per iteration. The optimization was performed using the Adam optimizer with a learning rate of $1 \times 10^{-4}$. All experiments were implemented in PyTorch 2.6.0 on a server running Ubuntu 22.04 equipped with an NVIDIA RTX 4090 GPU.

### 4.4. Quantitative and Visual Results

In Table 1, we present the quantitative comparison between our proposed method and existing state-of-the-art approaches on the Pavia University dataset. While the performance of the traditional Integrated method is not particularly remarkable, it remains stable across various distortion types. In contrast, MoGDCN, M2DTN, and IR&ArF exhibit high sensitivity to the specific type of distortion. RFDif performs poorly on the Pavia University dataset, yet as shown in the Appendix, it yields significantly better results on the Botswana dataset; PSRFDif demonstrates the opposite trend. This discrepancy suggests that these two diffusion-based architectures suffer from poor generalization. Furthermore, although the PRFCo method achieves competitive performance, it entails substantial computational overhead. Our method, designed based on a CNN architecture, maintains the lowest computational cost while achieving optimal performance due to the strategic design of the optimization objective. As illustrated in Figure 3, the reconstruction

errors for MoGDCN and IR&ArF are visually more pronounced. In comparison, our smaller difference indicates that CNN-based models, when coupled with precise loss function design, can achieve superior convergence. From Table 2 and Table 3, it can still be observed that our model achieves the best performance in most cases with the fewest parameters and floating-point operations. More analyses and visual results are provided in Appendix G.

### 4.5. Ablation and Parameter Sensitivity Studies

To validate the effectiveness of our specific design choices, we further conduct comprehensive ablation studies and sensitivity analyses based on the results obtained from the Pavia University dataset under elastic distortions.

**Impact of the two loss functions.** As illustrated in Table 4, the configuration "without $\mathcal{L}_{\text{RFM}}$" denotes the scenario where our objective degenerates into the standard velocity loss used in vanilla flow matching. The results indicate that while $\mathcal{L}_{\text{RFM}}$ exerts a more pronounced influence on performance, $\mathcal{L}_{\text{OTD}}$ also provides a discernible contribution. Crucially, both loss functions are shown to facilitate superior model convergence, thereby validating the effectiveness and necessity of our proposed optimization objectives.

**Impact of the sampling steps.** As demonstrated in Table 5, we compare our method with two representative diffusion-based approaches. The results indicate that our method achieves superior performance using significantly fewer sampling steps. While the sampling accuracy further improves as the number of steps increases, it concurrently introduces additional computational latency. Specifically,

*Table 3.* Quantitative comparison on the **Xiongan** dataset under Elastic, Homography, and Turbulence distortions. **Bold** indicates best, Underline indicates second-best. The gray column highlights our proposed method.

| DISTORTION | METRIC | Integrated (TGRS'20) | MoGDCN (TIP'21) | NonRegSR (TGRS'22) | M2DTN (AAAI'24) | MoEPNP (TNNLS'24) | IR&ArF (TIP'25) | PSRFDif (TGRS'25) | RFDif (TGRS'25) | PRFCo (TGRS'25) | Ours |
|---|---|---|---|---|---|---|---|---|---|---|---|
| Elastic ($\alpha$=600, $\sigma$=15) | PSNR↑ | 41.26 | 41.75 | 43.64 | 38.34 | 42.74 | 48.90 | 43.31 | 48.08 | 47.06 | **50.61** |
| | SSIM↑ | 0.981 | 0.986 | 0.989 | 0.980 | 0.986 | 0.997 | 0.991 | 0.996 | 0.996 | **0.997** |
| | SAM↓ | 2.455 | 2.359 | 1.959 | 2.943 | 2.272 | 1.149 | 2.040 | 1.246 | 1.400 | **0.999** |
| | ERGAS↓ | 1.112 | 1.047 | 0.843 | 1.551 | 0.937 | 0.462 | 0.878 | 0.509 | 0.570 | **0.381** |
| | RMSE↓ | 2.206 | 2.085 | 1.678 | 3.086 | 1.860 | 0.915 | 1.742 | 1.006 | 1.131 | **0.752** |
| Homography (MPO=10%) | PSNR↑ | 40.69 | 41.08 | 43.89 | 39.67 | 45.39 | 46.42 | 42.52 | 47.05 | 47.08 | **49.43** |
| | SSIM↑ | 0.972 | 0.988 | 0.989 | 0.981 | 0.993 | 0.995 | 0.991 | 0.994 | 0.996 | **0.997** |
| | SAM↓ | 2.557 | 2.458 | 1.940 | 2.786 | 1.696 | 1.416 | 2.063 | 1.371 | 1.377 | **1.083** |
| | ERGAS↓ | 1.186 | 1.127 | 0.820 | 1.331 | 0.692 | 0.615 | 0.960 | 0.571 | 0.568 | **0.435** |
| | RMSE↓ | 2.356 | 2.251 | 1.630 | 2.649 | 1.371 | 1.217 | 1.908 | 1.133 | 1.129 | **0.861** |
| Turbulence ($D/r_0$=2.0) | PSNR↑ | 41.63 | 46.33 | 44.00 | 40.67 | 40.26 | 49.62 | 43.75 | 50.15 | 48.07 | **51.98** |
| | SSIM↑ | 0.981 | 0.993 | 0.989 | 0.987 | 0.979 | 0.997 | 0.991 | 0.996 | 0.996 | **0.998** |
| | SAM↓ | 2.299 | 1.476 | 1.937 | 2.437 | 2.716 | 1.050 | 2.051 | 1.059 | 1.321 | **0.909** |
| | ERGAS↓ | 1.069 | 0.620 | 0.809 | 1.182 | 1.245 | 0.423 | 0.831 | 0.400 | 0.508 | **0.326** |
| | RMSE↓ | 2.114 | 1.230 | 1.609 | 2.360 | 2.474 | 0.843 | 1.655 | 0.793 | 1.007 | **0.642** |

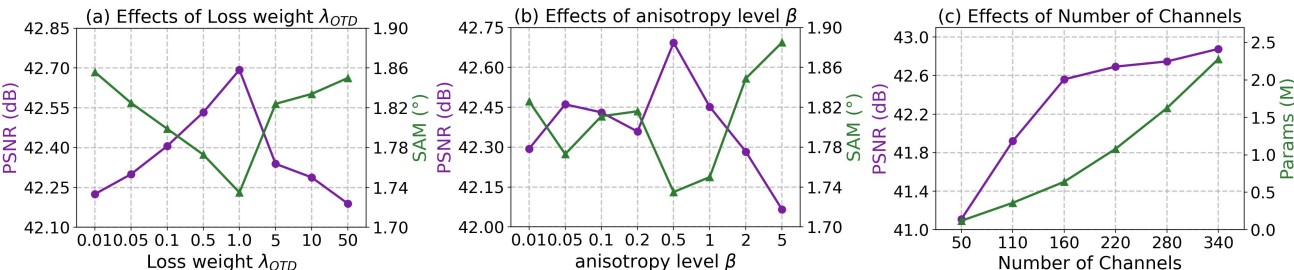

*Figure 4.* Parameter sensitivity analyses of $\lambda_{\mathrm{OTD}}$, $\beta$ and number of channels of hidden layers regarding the model performance.

*Table 4.* Ablation study of the proposed loss functions under elastic distortion on the Pavia University dataset. The gray row highlights our full model.

| $\mathcal{L}_{\mathrm{RFM}}$ | $\mathcal{L}_{\mathrm{OTD}}$ | PSNR↑ | SSIM↑ | SAM↓ | ERGAS↓ | RMSE↓ |
|---|---|---|---|---|---|---|
| × | × | 41.107 | 0.983 | 2.164 | 1.106 | 2.245 |
| ✓ | × | 42.142 | 0.986 | 1.884 | 0.986 | 1.993 |
| × | ✓ | 41.704 | **0.987** | 1.821 | 1.036 | 2.096 |
| ✓ | ✓ | **42.693** | **0.987** | **1.735** | **0.931** | **1.872** |

*Table 5.* Quantitative comparison and step analysis on the Pavia University dataset under elastic distortion. Units: Time (ms).

| METHOD | PSNR↑ | SSIM↑ | SAM↓ | ERGAS↓ | Time↓ | NFE↓ |
|---|---|---|---|---|---|---|
| PSRFDif | 41.492 | 0.982 | 2.015 | 1.067 | 33624 | 2000 |
| RFDif | 37.774 | 0.971 | 2.897 | 1.607 | 524.92 | 25 |
| Ours(20-step) | **42.897** | **0.988** | **1.689** | **0.908** | 117.43 | 20 |
| Ours(10-step) | 42.827 | **0.988** | 1.694 | 0.914 | 59.667 | 10 |
| Ours(5-step) | 42.734 | **0.988** | 1.693 | 0.925 | 28.126 | 5 |
| **Ours(1-step)** | 42.693 | 0.987 | 1.735 | 0.931 | **9.6574** | **1** |

the early PSRFDif requires 33,624 ms for a full sampling cycle, whereas our one-step implementation takes only 9.66 ms. Combined with the significantly lower parameter count and FLOPs reported in Table 2, our method is far more conducive to practical real-world deployment.

**Construction of the Riemannian metric $M$ in Eq. (7).** We conduct ablation experiments on the specific configurations of $M$, and the results are presented in Table 6. Specifically, the first row represents the baseline scenario where optimization is performed entirely in Euclidean space. The second row evaluates a variation where gradient information is obtained by taking the maximum gradient across all bands, rather than the band-by-band accumulation typically

used for HR-MSI. The third row corresponds to utilizing the average gradient across all spectral bands. In the fourth row, the model incorporates gradient information via isotropic modeling, injecting it into the network as a standard feature map. The fifth row involves replacing the Sobel edge detection operator with the more sophisticated Scharr operator, while the sixth row employs the simpler Prewitt operator. Our analysis indicates that the utilization of gradients significantly enhances model performance, which is consistent with our empirical observations. Furthermore, the specific choice of gradient operators and the method of utilizing gradient information have a discernible but minor impact on performance. Finally, the selection between Riemannian and Euclidean space affects performance in a manner com-

*Table 6.* Ablation study of the Riemannian metric construction $\mathbf{M}$ on the Pavia University dataset under elastic distortion.

| METRIC FORMULATION | PSNR↑ | SSIM↑ | SAM↓ | ERGAS↓ | RMSE↓ |
|---|---|---|---|---|---|
| $\mathbf{M} = \mathbf{I}$ (Euclidean) | 41.933 | 0.987 | 1.782 | 1.010 | 2.041 |
| maximum $\max_c(\mathbf{I}_c)$ | 42.256 | 0.987 | 1.768 | 0.975 | 1.967 |
| average $\text{avg}_c(\mathbf{I}_c)$ | 42.323 | 0.987 | 1.777 | 0.967 | 1.952 |
| Isotropic Structure Metric | 42.056 | 0.987 | 1.774 | 0.997 | 2.013 |
| Scharr Operator | 42.564 | 0.987 | 1.749 | 0.942 | 1.898 |
| Prewitt Operator | 42.294 | 0.987 | 1.770 | 0.971 | 1.958 |
| **Ours** | **42.693** | 0.987 | **1.735** | **0.931** | **1.872** |

*Table 7.* Ablation study on different numerical approximations of the Riemannian transport cost on the Pavia University dataset under elastic distortion. The gray row highlights our selected strategy.

| $c(u, u+\Delta u)$ | PSNR↑ | SAM↓ | ERGAS↓ | Time (ms) | FLOPs (M) |
|---|---|---|---|---|---|
| $\Delta u^\top M\left(\frac{u+(u+\Delta u)}{2}\right)\Delta u$ | 42.821 | 1.716 | 0.914 | 4.6551 | 9.6338 |
| $\sum_{i=0}^{L-1}\Delta u_i^\top M(x_i)\Delta u_i$ | **42.918** | **1.707** | **0.904** | 13.856 | 32.440 |
| $\Delta u^\top M(u)\Delta u$ | 42.693 | 1.735 | 0.931 | **4.2495** | **6.4881** |

*Table 8.* Ablation study on different backbone networks on the Pavia University dataset under elastic distortion.

| BACKBONE | PSNR↑ | SSIM↑ | SAM↓ | ERGAS↓ | Params (M)↓ | FLOPs (G)↓ |
|---|---|---|---|---|---|---|
| ResNet-18 | 41.286 | 0.985 | 1.933 | 1.065 | 14.53 | 18.10 |
| U-Net | 39.849 | 0.981 | 2.380 | 1.275 | 31.51 | **16.03** |
| ViT-Base | **43.800** | **0.990** | **1.563** | **0.826** | 130.5 | 56.27 |
| **Ours** | 42.693 | 0.987 | 1.735 | 0.931 | **1.078** | 21.29 |

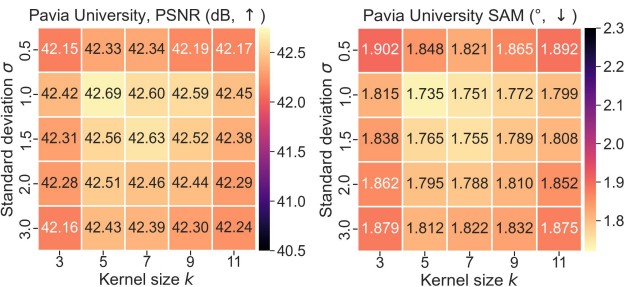

*Figure 5.* Parameter sensitivity analyses of standard deviation $\sigma$ and kernel size $k$ of Gaussian kernel.

parable to the influence of the isotropic modeling approach.

**Approximations of the Riemannian Transport Cost.** We conducted ablation experiments on the numerical approximations for Eq. (11), with results summarized in Table 7. The first row evaluates the midpoint approximation, which computes the Riemannian metric at the trajectory's center. The second row represents the discrete path integral, which accumulates transport costs along the entire trajectory. In contrast, our approach freezes the metric at the source location; this strategy achieves competitive performance—particularly in spectral mapping (SAM)—while significantly reducing computational overhead in terms of both FLOPs and inference time.

**Backbone Networks.** Given our focus on Riemannian space modeling and objective optimization, we evaluate different velocity and fusion network backbones in Table 8. Our CNN-based choice provides an optimal trade-off between efficiency and performance. While ViT architectures can offer superior accuracy, they entail significant computational overhead, justifying our selection for efficient deployment.

**Parameter Sensitivity.** Comprehensive sensitivity analyses were conducted on the balancing weight $\lambda_{OTD}$, anisotropy intensity $\beta$, and hidden layer channels (shown in Figure 4). Results indicate that performance follows an inverted U-curve with respect to $\lambda_{OTD}$, suggesting that an excessive weight dominates and hinders the optimization of the primary loss $\mathcal{L}_{\text{RFM}}$. The best weight is $\lambda_{OTD} = 1.0$. Regarding anisotropy, excessively high $\beta$ values cause extreme spatial distortion, leading to performance degradation. We select $\beta = 0.5$ in our paper. While increasing the channel count improves performance, the gains eventually saturate as the parameter volume grows; we thus select $channels = 220$. Additionally, ablation studies on the Gaussian blur kernel in Eq. (6) as shown in Figure 5 reveal that performance suffers when boundaries are overly blurred or when the blurring effect is insufficient. We finally set the kernel size to $k = 5$ and the standard deviation to $\sigma = 1.0$.

## 5. Conclusion

In this paper, we address the challenge of joint registration and fusion of hyperspectral and multispectral images under complex non-rigid distortions from the perspective of Riemannian manifold modeling. We propose Structure-Aware Riemannian Flow Matching (SA-RFM), a geometry-informed framework that reformulates image registration as dynamic optimal transport on a structure-induced manifold. SA-RFM integrates anisotropic, structure-dependent transport costs into a conditional flow matching formulation through a local cost approximation, and further introduces an optimal transport direction regularization to align the learned deformation field with minimum-cost transport directions under the induced geometry. Extensive experiments on multiple benchmarks under different distortions demonstrate that SA-RFM consistently outperforms existing methods in terms of accuracy and computational efficiency.

## Acknowledgements

This work was supported in part by the Young Scientists Fund of the National Natural Science Foundation of China under No. 62401578 and in part by the Shandong Provincial Natural Science Foundation under ZR2023LZH014.

## Impact Statement

This paper presents work whose goal is to advance the field of Machine Learning. There are many potential societal consequences of our work, none which we feel must be specifically highlighted here.

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

# A. Appendix Outline

In this supplementary material, we provide more details and comprehensive experimental results of our proposed Structure-Aware Riemannian Flow Matching (SA-RFM) framework as follows:

- Sec. B presents more theoretical derivations and proofs.

- Sec. C provides a comprehensive review of Related Work, covering traditional paradigms, deep learning-based approaches, and the theoretical foundations of Riemannian one-step flow models.

- Sec. D introduces the benchmarks used for comparison in our experiments.

- Sec. E defines the fusion metrics employed to quantitatively assess the registration and fusion performance.

- Sec. F describes the details of the datasets.

- Sec. G reports more quantitative and visualization results to further demonstrate the generalization capability and robustness of our method.

- Sec. H presents robustness experiments under realistic degradation scenarios, such as noise, artifacts, and spectral variations.

- Sec. I conducts a theoretical complexity analysis of the proposed network, including both time and space complexity.

# B. Additional theoretical supplements and explanations

### B.1. Derivation of Eq. (9).

Eq. (9) is obtained by combining the standard dynamic optimal transport formulation in Eq. (2) with the structure-induced Riemannian metric defined in Eq. (7). Specifically, the standard dynamic optimal transport objective minimizes the Euclidean kinetic energy:

$$\inf_{v_t} \int_0^1 \int_\Omega \frac{1}{2} \|v_t(u)\|_2^2 \, \rho_t(u) \, du \, dt. \tag{17}$$

To incorporate the structural prior from the reference image, we equip the spatial domain $\Omega$ with a position-dependent Riemannian metric tensor:

$$M(u) = I + \beta S_Y(u). \tag{18}$$

Accordingly, the Euclidean norm of the velocity field is replaced by the squared Riemannian norm:

$$\|v_t(u)\|_{M(u)}^2 = v_t(u)^\top M(u) v_t(u). \tag{19}$$

Substituting this metric-induced kinetic energy into the dynamic optimal transport objective gives:

$$\mathcal{J}(v_t) = \inf_{v_t} \int_0^1 \int_\Omega \frac{1}{2} v_t(u)^\top M(u) v_t(u) \rho_t(u) \, du \, dt. \tag{20}$$

This structure-aware formulation penalizes transport across strong structural boundaries, where $M(u)$ has large eigenvalues, and thus encourages the learned transport trajectory to align with the underlying image geometry.

### B.2. Proof of the closed-form solution for Eq. (13) and Eq. (14)

Eq. (13) is a quadratic program with a linear constraint:

$$\min_v v^\top M(u)v, \quad \text{s.t.} \quad d(u)^\top v = \|d(u)\|_2. \tag{21}$$

We construct the Lagrangian $\mathcal{L}(v, \lambda)$ as

$$\mathcal{L}(v, \lambda) = v^\top M(u)v + \lambda \left( \|d(u)\|_2 - d(u)^\top v \right). \tag{22}$$

Setting its gradient with respect to $v$ to zero gives

$$\nabla_v \mathcal{L} = 2M(u)v - \lambda d(u) = 0. \tag{23}$$

Using the invertibility of $M(u)$, we obtain

$$v = \frac{\lambda}{2} M(u)^{-1} d(u). \tag{24}$$

Substituting this expression for $v$ into the original constraint $d(u)^\top v = \|d(u)\|_2$ allows us to solve for the scalar $\lambda/2$:

$$\frac{\lambda}{2} \left( d(u)^\top M(u)^{-1} d(u) \right) = \|d(u)\|_2 \quad \Longrightarrow \quad \frac{\lambda}{2} = \frac{\|d(u)\|_2}{d(u)^\top M(u)^{-1} d(u)}. \tag{25}$$

Plugging $\lambda/2$ back into the expression for $v$ yields the exact closed-form solution (Eq. (14)):

$$v^\dagger(u) = \frac{M(u)^{-1} d(u)}{d(u)^\top M(u)^{-1} d(u)} \|d(u)\|_2. \tag{26}$$

### B.3. Rationale for Using OTD Regularization Instead of Fully Riemannian Flow Matching

While a fully Riemannian Flow Matching (RFM) framework following exact geodesics is mathematically elegant, its application to high-dimensional image registration faces a severe computational bottleneck.

Calculating exact geodesics requires solving the second-order ODE

$$\ddot{\gamma}^k + \Gamma_{ij}^k \dot{\gamma}^i \dot{\gamma}^j = 0, \tag{27}$$

which depends on the Christoffel symbols $\Gamma_{ij}^k$ derived from the spatial derivatives of the metric $M(u)$. Since our structure-induced metric is highly data-dependent and derived from discrete high-resolution gradients, solving this global Boundary Value Problem (BVP) across millions of pixels at every training iteration is computationally intractable.

Our OTD regularization is explicitly designed as a computationally efficient surrogate for these geodesics. Instead of an expensive global BVP, OTD performs a local, first-order geometric correction. By analytically solving Eq. (13), the resulting $v^\dagger(u)$ acts as a tangent approximation: it maintains the global Euclidean direction to ensure convergence, while its local trajectory is warped by $M(u)^{-1}$ to favor minimum-cost structural paths.

### B.4. Exact Formulation of the Structure Tensor

We compute the MSI grayscale gradients $\nabla I = [I_x, I_y]^\top$ and the base tensor

$$J_0 = \nabla I \nabla I^\top. \tag{28}$$

We extract its principal eigenvalue $\lambda_1$ and unit eigenvector $\mathbf{e}_1 = [e_x, e_y]^\top$, with the orthogonal vector $\mathbf{e}_2 = [-e_y, e_x]^\top$.

To penalize transport across strong edges, we non-linearly amplify the principal eigenvalue as

$$\lambda_1' = 1 + \alpha \lambda_1^k, \tag{29}$$

where $\alpha = 10$ and $k = 2$, while keeping $\lambda_2' = 1$. The exact metric tensor used in our code is:

$$M(u) = \lambda_1' \mathbf{e}_1 \mathbf{e}_1^\top + \mathbf{e}_2 \mathbf{e}_2^\top = \begin{bmatrix} \lambda_1' e_x^2 + e_y^2 & (\lambda_1' - 1) e_x e_y \\ (\lambda_1' - 1) e_x e_y & \lambda_1' e_y^2 + e_x^2 \end{bmatrix}. \tag{30}$$

### B.5. Theoretical bounds and approximation error for the simplified Riemannian energy

By freezing the metric at $u$ and assuming a straight-line path for displacement $\Delta u$, we perform a first-order Taylor approximation of the Riemannian energy. The approximation error is bounded by:

$$\text{Error} \leq \mathcal{O}\left( \|\nabla M(u)\| \|\Delta u\|^3 \right). \tag{31}$$

Since $M(u) = I + \beta S_Y(u)$ (Eq. (7)), this bound is directly constrained by the smoothness, i.e., the Hessian, of the multispectral image structure tensor $S_Y(u)$.

A straight-line assumption for large, single-step displacements may cause the approximation error to blow up in other approaches. However, our approach avoids this by operating within a continuous dynamic Flow Matching ODE paradigm. Rather than a single static leap, the total deformation is governed by

$$\frac{ds_t}{dt} = v_t(s_t). \tag{32}$$

During inference, the ODE solver breaks the global deformation into discrete, infinitesimal integration steps $dt$, where the instantaneous displacement is exactly

$$\Delta u = v_t dt. \tag{33}$$

Since a small $dt$ keeps the local displacement $\|\Delta u\|$ minimal, our quadratic approximation becomes asymptotically exact as $dt \to 0$. Thus, the integration process prevents prohibitive error accumulation even under extreme macroscopic deformations. Furthermore, performance can be improved by increasing the number of steps, rather than being restricted to one-step inference.

## C. Related Work

### C.1. Traditional Paradigms for Hyperspectral and Multispectral Image Registration

Traditionally, the fusion of hyperspectral images (HSIs) and multispectral images (MSIs) has predominantly relied on a two-stage paradigm, where geometric alignment is treated as a prerequisite preprocessing step decoupled from the subsequent fusion task(Ying et al., 2022; Zhou et al., 2020). To address the spatial misalignment between LR-HSIs and HR-MSIs, early approaches largely relied on feature-based techniques, extracting distinct keypoints via Scale-Invariant Fourier Transform (SIFT) or Harris detectors, and subsequently approximating non-rigid geometric distortions through Thin-Plate Splines (TPS) or Gaussian radial basis functions. Parallelly, area-based strategies have been employed to optimize similarity metrics across modalities; for instance, Zhou et al. proposed minimizing a least-squares objective function based on the Point Spread Function (PSF)(Zhou et al., 2017), while Ying et al. introduced the Normalized Edge Difference (NED) to align images by minimizing edge discrepancies(Ying et al., 2022). However, this decoupled strategy ignores the intrinsic correlation between registration and fusion, inevitably leading to error propagation where misregistration artifacts cannot be rectified during the fusion stage. Consequently, these conventional methods exhibit limited robustness when handling complex non-rigid deformations induced by sensor dynamics or terrain irregularities in remote sensing scenarios.

### C.2. Deep Learning-Based Joint Registration and Fusion for Hyperspectral and Multispectral Images

The advent of deep learning has precipitated a paradigmatic shift in HSI processing from independent preprocessing steps to end-to-end joint optimization frameworks. Initial efforts adapted Spatial Transformer Networks (STN)(Jaderberg et al., 2015) and unsupervised deformation fields akin to VoxelMorph(Balakrishnan et al., 2019) to estimate affine or non-rigid transformations. To circumvent the "error accumulation" inherent in cascaded pipelines, diverse strategies have emerged. For instance, u2-MDN(Qu et al., 2021) circumvents explicit registration by disentangling the feature extraction process, specifically deriving spectral representations from HSIs and spatial details from MSIs, respectively. However, the mainstream has evolved towards unified architectures where registration and fusion are mutually reinforcing. Zheng et al. pioneered this with NonRegSRNet, integrating dense registration and fusion into a unified CNN-based model(Zheng et al., 2021). While effective, CNN-based methods are often constrained by local receptive fields. Addressing this, Transformer-based architectures have been introduced to capture long-range dependencies, such as SCANet which simulates stereo-vision processes(Guo et al., 2023), and M2DTN which leverages deformable attention to align multi-scale features(Dong et al., 2024). Furthermore, to mitigate the computational burden of self-attention mechanisms, recent studies have leveraged Mamba-based architectures to achieve efficient progressive registration-fusion co-optimization(Li et al., 2025). Parallel to these architecture innovations, model-driven networks have gained prominence by mathematically unfolding the physical registration-fusion process into deep layers, thereby enhancing interpretability. Notable approaches include integrating physical perception models for registration-fusion consistency(Qu et al., 2025a) or constructing arbitrary-resolution observation models(Qu et al., 2025c). The aforementioned schemes are typically two-stage, assigning the registration and fusion tasks to separate networks; consequently, they are prone to introducing cross-task errors during the

optimization process. Most recently, Generative AI has also been adapted for this task, where diffusion models are employed to model the joint distribution of registration and fusion in a probabilistic manner(Qu et al., 2025b; Diao et al., 2025).

### C.3. Riemannian Manifold Learning and One-Step Flow Generative Models

While diffusion models have shown promise in joint distribution modeling, their reliance on flat Euclidean metrics and computationally intensive iterative sampling necessitates a shift toward geometry-aware and efficient generative paradigms. From a geometric standpoint, hyperspectral data intrinsically resides on non-linear manifolds where Euclidean measures fail to capture geodesic distances essential for accurate deformation modeling. This has motivated the adoption of Riemannian geometry, tracing back to variational methods like LDDMM(Beg et al., 2005) that ensure topological preservation, and evolving into modern geometric deep learning that learns structure-aware metric tensors to guide image restoration(Arvanitidis et al., 2017). Complementing this geometric perspective, Flow Matching (FM) and Optimal Transport (OT) have emerged to address the sampling inefficiency of standard diffusion. Unlike noise-driven trajectories, FM constructs deterministic Ordinary Differential Equation (ODE) paths that minimize transport costs(Lipman et al., 2022; Tong et al., 2023).

To further bridge the gap between high-fidelity generation and real-time inference, recent advancements in one-step generation, exemplified by Rectified Flow(Liu et al., 2022) and Consistency Models(Song et al., 2023), have enabled the distillation of complex ODE trajectories into a single forward pass. These methods straighten the transport paths in the probability space, allowing for efficient coupling between source and target distributions. However, existing one-step frameworks predominantly operate under isotropic Euclidean assumptions. Integrating Riemannian metric constraints into one-step flow matching for the specific challenge of unregistered HSI-MSI fusion remains an underexplored frontier, promising to combine the geometric fidelity of manifold learning with the computational efficiency of rectified flows.

## D. The Benchmarks

**Integrated** proposes a joint optimization framework where registration utilizes B-spline free-form deformation and fusion is solved via a closed-form solution derived from a manifold-invariant assumption, effectively handling rigid and non-rigid distortions.

**MoGDCN** unfolds an iterative optimization algorithm into a model-guided deep network. By explicitly integrating the HSI observation model and a learned spatial-spectral denoiser, it combines the interpretability of model-based methods with the learning capability of CNNs to reconstruct HR-HSI.

**NonRegSR** introduces a novel unsupervised framework composed of a spatial transformation network and a triplet architecture. It simultaneously performs non-rigid registration and spectral unmixing-based super-resolution without relying on ground-truth labels, using a multitask learning strategy to ensure the two tasks complement each other.

**M2DTN** addresses the limitation of CNNs in capturing global features by proposing a Multi-Modal Cross-Scale Deformable Transformer. Built upon a Spectrum-Preserving Spatial-Guided Registration-SR Unified (SSRU) model, it leverages deformable cross-attention to align features across scales, effectively correcting distortions while preserving spectral fidelity in unregistered scenarios.

**MoEPNP** presents a theoretically grounded approach by designing a Registration-Fusion Consistency Model (RFCM) that mathematically couples the two tasks to reduce sensitivity to registration errors. To solve this, it employs a Mixture-of-Experts (MoE) Plug-and-Play framework, where an MoE-based hyper-solver dynamically adjusts parameters, ensuring high interpretability and robust performance against misalignments.

**IR&ArF** targets the challenging problem of arbitrary resolution fusion. It constructs a Spatial-Spectral Consistent Arbitrary Scale Observation Model and unfolds its optimization into a deep interpretable network. This design allows for model-data dual-driven fusion, enabling the generation of high-quality HSIs at continuous resolution scales even under unregistered conditions.

**PSRFDif** leverages the generative power of diffusion models through a Progressive Synergistic Registration and Fusion Diffusion Network. It introduces a Fine Registration and Fusion Bridging (FRFB) model to guide the reverse diffusion process, iteratively co-optimizing registration and fusion to handle complex, non-linear misalignments that traditional methods fail to capture.

**RFDif** integrates registration and fusion into a unified dual-conditional diffusion framework. It establishes a closed-loop mechanism where the registration module corrects misalignment to guide fusion, and the fused result feedbacks to refine registration, allowing the two tasks to mutually enhance each other iteratively.

**PRFCo** explores the efficiency of State Space Models by utilizing an A-Mamba network. It employs a progressive co-optimization strategy with a Modal-Aligned Progressive Registration-Fusion (MAPRF) module, which corrects deformations from coarse to fine levels while refining fusion results, offering a computationally efficient alternative to Transformer-based methods.

## E. Fusion Metrics

Four widely used fusion metrics are used to evaluate the performance of each method.

(1) Peak Signal to Noise Ratio: The PSNR is one of the most commonly used indicators to measure image quality (Hore & Ziou, 2010):

$$\text{PSNR} = 10 \cdot \log_{10} \left( \frac{\max \left( \mathbf{R}_c \right)^2}{\frac{1}{HW} \left\| \mathbf{R}_c - \mathbf{Z}_c \right\|_2^2} \right) \tag{34}$$

where $\mathbf{R}_c$ and $\mathbf{Z}_c$ represent the $c$th channel of the reference image and the predicted image, respectively. $\|\cdot\|_2$ means the second norm. After adding up the PSNRs of all channels, we obtain the PSNR metric of the whole image. The larger the PSNR, the better the performance.

(2) Structural Similarity Index Measure: The SSIM is an indicator used to measure the degree of similarity between two digital images (Wang et al., 2004). Structural similarity aligns more closely with the human visual perception of image quality, providing a metric that better reflects the discernment of human eyes. The SSIM is formulated as

$$\text{SSIM} = [l(\mathbf{R}, \mathbf{Z})]^\alpha \cdot [c(\mathbf{R}, \mathbf{Z})]^\beta \cdot [s(\mathbf{R}, \mathbf{Z})]^\gamma \tag{35}$$

$$l(\mathbf{R}, \mathbf{Z}) = \frac{2\mu_\mathbf{R}\mu_\mathbf{Z} + C_1}{\mu_\mathbf{R}^2 + \mu_\mathbf{Z}^2 + C_1} \tag{36}$$

$$c(\mathbf{R}, \mathbf{Z}) = \frac{2\sigma_\mathbf{R}\sigma_\mathbf{Z} + C_2}{\sigma_\mathbf{R}^2 + \sigma_\mathbf{Z}^2 + C_2} \tag{37}$$

$$s(\mathbf{R}, \mathbf{Z}) = \frac{\sigma_{\mathbf{R}\mathbf{Z}} + C_3}{\sigma_\mathbf{R}\sigma_\mathbf{Z} + C_3} \tag{38}$$

where $l$, $c$, and $s$ represent a comparison of the brightness, contrast, and structure between the reference image and the predicted image, respectively. $\alpha, \beta, \gamma$ are weighting coefficients, typically set to 1. $\mu_\mathbf{R}$ and $\mu_\mathbf{Z}$ denote the average value of $\mathbf{R}$ and $\mathbf{Z}$, while $\sigma_\mathbf{R}$ and $\sigma_\mathbf{Z}$ denote the standard deviation. $\sigma_{\mathbf{R}\mathbf{Z}}$ denotes the covariance of $\mathbf{R}$ and $\mathbf{Z}$. The higher the SSIM, the better the performance.

(3) Spectral Angle Mapper: The SAM is a spectral angle mapping that treats the spectrum of each pixel in an image as a high-dimensional vector (Yuhas et al., 1992). The similarity between spectra is measured by calculating the angle between the two vectors. The SAM is defined as

$$\text{SAM} = \cos^{-1} \left( \frac{\sum_{c=1}^{C} \mathbf{r}_c \mathbf{z}_c}{\sqrt{\sum_{c=1}^{C} \mathbf{r}_c^2} \sqrt{\sum_{c=1}^{C} \mathbf{z}_c^2}} \right) \tag{39}$$

where $\mathbf{r}_c$ and $\mathbf{z}_c$ refer to the spectral vectors of the reference and predicted HR-HSIs. The smaller the SAM, the more similar the two spectra.

(4) Relative Dimensionless Global Error in Synthesis: ERGAS is an indicator used to evaluate the quality of remote sensing images (Wald, 2000); it considers the mean square error and brightness information of the image to provide a comprehensive evaluation of the performance of the image. The definition formula is as follows:

$$\text{ERGAS} = \frac{100}{r} \sqrt{\frac{1}{C} \sum_{c=1}^{C} \frac{\left\| \mathbf{R}_c - \mathbf{Z}_c \right\|_2^2}{\mu^2 \left( \mathbf{R}_c \right)}} \tag{40}$$

where $r$ refers to the spatial downsampling ratio when we obtain the LR-HSI. $\mu\left(\mathbf{R}_c\right)$ represents the mean value of the $c$th channel of the reference image. The lower the ERGAS value, the higher the image quality.

(5) Mean Relative Absolute Error: The MRAE measures the relative deviation between the reconstructed image and the ground truth, providing an assessment of spectral fidelity relative to the signal intensity (Arad & Ben-Shahar, 2016). It is defined as the mean of the absolute differences normalized by the ground truth values. The MRAE is formulated as

$$\mathrm{MRAE} = \frac{1}{N}\sum_{i=1}^{N}\frac{|\mathbf{R}_i - \mathbf{Z}_i|}{\mathbf{R}_i} \tag{41}$$

where $N$ denotes the total number of pixels in the image. $\mathbf{R}_i$ and $\mathbf{Z}_i$ represent the spectral value of the $i$-th pixel in the reference image and the predicted image, respectively. The smaller the MRAE, the better the performance. This metric is only used in visual results.

## F. Datasets

Here, we provide detailed descriptions of the datasets used in the main text, as well as those used in the supplementary experiments. All the datasets can be downloaded online.

(1) Pavia University: The Pavia University dataset was captured by ROSIS sensors over Pavia University in Italy. After removing samples without any information, the dataset consists of $610 \times 340$ pixels and 103 spectral bands, with wavelengths ranging from 0.43 to 0.86 $\mu$m at a spatial resolution of 1.3 m.

(2) Botswana: The Botswana dataset contains a sequence of data captured by the Hyperion sensor on the EO-1 satellite over Okavango Delta, Botswana. After removing bands that cover the water absorption characteristics, the dataset consists of $1476 \times 256$ pixels and 145 bands, with spectral wavelengths ranging from 0.4 to 2.5 $\mu$m at a spatial resolution of 30 m.

(3) Chikusei: The Chikusei dataset was acquired by the Headwall Hyperspec-VNIR-C imaging sensor over the agricultural and urban areas of Chikusei, Ibaraki, Japan. This dataset features a large scene size of $2517 \times 2335$ pixels with a spatial resolution of 2.5-m. It covers 128 spectral bands across the wavelength range of 0.36–1.02 $\mu$m.

(4) Xiongan: The Xiongan dataset was captured by an airborne sensor over the Xiongan New Area, China. This high-resolution dataset includes $3750 \times 1580$ pixels with a fine spatial resolution of 0.5-m. It spans 256 spectral bands ranging from 0.39 to 1.00 $\mu$m.

## G. Additional Quantitative and Visual Results of Comparative Experiments

As the detailed analysis of the Pavia University dataset is presented in the main manuscript, this section focuses on the evaluations of the Botswana, Chikusei, and Xiongan datasets under three distinct deformation types: Elastic, Homography, and Turbulence. Table 2, Table 9, and Table 3 present the comprehensive results, where the best and second-best results are highlighted in boldface and underlined, respectively. In the quantitative analysis results, the traditional method Integrated exhibits notable stability across varying scenarios despite not achieving top-tier numerical scores. Among deep learning-based approaches, IR&ArF demonstrates consistently favorable performance across all three datasets. However, generative models exhibit inconsistent performance; specifically, RFDif performs exceptionally well on the Botswana and Xiongan datasets but degrades noticeably on the Chikusei dataset, indicating insufficient robustness and limited generalization capability. Conversely, PSRFDif, while delivering less outstanding peak performance, maintains a more stable output compared to RFDif. Furthermore, while PRFCo achieves competitive restoration quality, it incurs a substantial computational burden. In sharp contrast, our proposed framework, designed based on a lightweight CNN architecture, requires the minimal computational overhead (1.203 M parameters). However, attributed to our carefully designed optimization objective, it achieves optimal outcomes across almost all metrics (e.g., PSNR, SSIM, and SAM), confirming its superior efficiency and generalization capability compared to both traditional and state-of-the-art deep learning-based methods.

Additionally, we conduct qualitative analysis to visually assess the registration and fusion quality. The visual comparisons on the Pavia University, Botswana, Chikusei, and Xiongan datasets are illustrated in Figures 7, 8, 9, and 10. From the visual comparisons, particularly on the Pavia University and Botswana datasets, it is evident that competitive methods either suffer severe ghosting artifacts caused by misregistration, such as Integrated and NonRegSR, or exhibit pronounced spectral distortions, as seen with PSRFDif and RFDif. On the Chikusei and Xiongan datasets, even though the visual disparities

*Table 9.* Quantitative comparison on the **Chikusei** dataset under Elastic, Homography, and Turbulence distortions. **Bold** indicates best, Underline indicates second-best. The gray column highlights our proposed method.

| DISTORTION | METRIC | Integrated (TGRS'20) | MoGDCN (TIP'21) | NonRegSR (TGRS'22) | M2DTN (AAAI'24) | MoEPNP (TNNLS'24) | IR&ArF (TIP'25) | PSRFDif (TGRS'25) | RFDif (TGRS'25) | PRFCo (TGRS'25) | Ours |
|---|---|---|---|---|---|---|---|---|---|---|---|
| Elastic ($\alpha$=600, $\sigma$=15) | PSNR↑ | 38.65 | 38.74 | 38.25 | 38.54 | 39.74 | 40.77 | 38.28 | 39.57 | 39.74 | **41.77** |
| | SSIM↑ | 0.982 | 0.983 | 0.980 | 0.983 | 0.987 | 0.989 | 0.983 | 0.988 | 0.988 | **0.991** |
| | SAM↓ | 3.157 | 3.282 | 3.555 | 3.171 | 2.549 | 2.311 | 3.320 | 2.494 | 2.378 | **2.153** |
| | ERGAS↓ | 1.378 | 1.365 | 1.438 | 1.388 | 1.208 | 1.076 | 1.436 | 1.230 | 1.208 | **0.962** |
| | RMSE↓ | 2.979 | 2.948 | 3.118 | 3.017 | 2.627 | 2.334 | 3.107 | 2.680 | 2.626 | **2.080** |
| Homography (MPO=10%) | PSNR↑ | 38.34 | 38.21 | 38.58 | 37.02 | 39.42 | 38.47 | 38.53 | 39.54 | **42.33** | 41.21 |
| | SSIM↑ | 0.982 | 0.983 | 0.982 | 0.975 | 0.986 | 0.983 | 0.983 | 0.988 | 0.983 | **0.991** |
| | SAM↓ | 3.158 | 3.352 | 3.379 | 4.143 | 2.759 | 3.189 | 3.292 | 2.515 | **1.758** | 2.232 |
| | ERGAS↓ | 1.427 | 1.444 | 1.387 | 1.652 | 1.256 | 1.398 | 1.396 | 1.234 | **0.972** | 1.026 |
| | RMSE↓ | 3.087 | 3.134 | 3.003 | 3.595 | 2.727 | 3.042 | 3.021 | 2.689 | **1.951** | 2.218 |
| Turbulence ($D/r_0$=2.0) | PSNR↑ | 40.20 | 40.50 | 39.70 | 39.71 | 41.07 | 42.74 | 39.38 | 42.79 | 41.13 | **43.61** |
| | SSIM↑ | 0.984 | 0.985 | 0.983 | 0.986 | 0.987 | 0.991 | 0.984 | 0.991 | 0.989 | **0.993** |
| | SAM↓ | 2.846 | 2.855 | 3.196 | 2.742 | 2.506 | 2.045 | 3.140 | 1.964 | 2.205 | **1.735** |
| | ERGAS↓ | 1.156 | 1.110 | 1.221 | 1.209 | 1.042 | 0.867 | 1.267 | 0.858 | 1.034 | **0.782** |
| | RMSE↓ | 2.492 | 2.408 | 2.641 | 2.636 | 2.256 | 1.860 | 2.740 | 1.849 | 2.239 | **1.683** |

*Table 10.* Performance evaluation and sensitivity analysis on the Pavia University dataset under elastic distortion. We evaluate the impact of spectral variations and occlusions in LR-HSI, alongside various MSI-side distortions such as additive noise, spatial misregistration, and imaging artifacts.

| Setting | PSNR ↑ | SSIM ↑ | SAM ↓ | ERGAS ↓ | RMSE ↓ | Experimental Description |
|---|---|---|---|---|---|---|
| Ours (Baseline) | 42.69 | 0.987 | 1.735 | 0.931 | 1.872 | Standard Elastic ($\alpha = 600, \sigma = 15$) |
| Registered LR-HSI | 42.97 | 0.988 | 1.689 | 0.898 | 1.811 | Results of undistorted LR-HSI and HR-MSI |
| Large Distortion | 42.45 | 0.987 | 1.773 | 0.955 | 1.924 | Strong Elastic ($\alpha = 1000, \sigma = 15$) |
| HSI Occlusion | 41.40 | 0.986 | 1.933 | 1.066 | 2.169 | Robustness against partial HSI occlusion |
| HSI Spectral Variation | 38.59 | 0.958 | 2.037 | 1.491 | 3.000 | Impact of HSI spectral signatures variation |
| MSI Noise Injection | 40.97 | 0.982 | 2.221 | 1.124 | 2.281 | MSI with additive Gaussian noise |
| MSI Misalignment | 28.69 | 0.830 | 9.536 | 4.526 | 9.375 | Effect of spatial misregistration in MSI |
| MSI Artifact | 36.72 | 0.965 | 3.604 | 1.813 | 3.719 | Effect of imaging artifacts in MSI |

between state-of-the-art methods and ours appear relatively subtle, our method maintains a distinct advantage in quantitative fidelity. Compared with these CNN and Diffusion-based methods, our method models the deformation flow based on structure-induced Riemannian geometry, which offers significant advantages in respecting object boundaries and ensuring pixel-level accuracy during transport. Overall, our method demonstrates outstanding performance in both quantitative and qualitative experiments across diverse datasets, indicating its strong generalizability.

## H. Robustness Experiments on LR-HSI and HR-MSI

In practical applications, LR-HSI and HR-MSI are often affected by various environmental noises, artifacts, and spectral distortions. To evaluate the robustness of our method under these realistic challenges, we design a series of degradation experiments on the Pavia University dataset, targeting both LR-HSI and HR-MSI. The visual results are shown in Figure 6, and the quantitative results are reported in Table 10. In the table, we additionally include a baseline experiment conducted on the already registered LR-HSI. It can be observed that the fusion performance is the best when no degradation is introduced.

Moreover, the fourth row of Table 10 shows the influence of stronger elastic distortion. The results indicate that, as the distortion level increases, the degradation in fusion performance is not significant. Notably, we observe that introducing

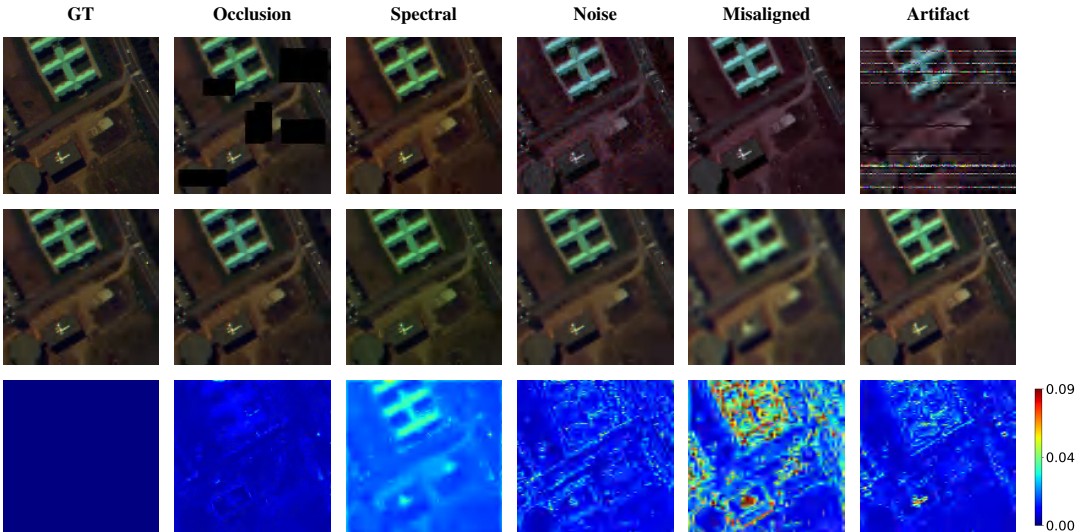

*Figure 6.* Visual comparison of the fusion results on the Pavia University dataset under various degradation scenarios before elastic distortion ($\alpha = 600, \sigma = 15$). The top row illustrates the simulated low-quality inputs: **Occlusion** and **Spectral** variations are applied to the original LR-HSI, while **Noise**, **Misaligned**, and **Artifact** interferences are introduced to the MSI. (Note that elastic distortion is applied after degradations.) The middle row displays the reconstructed HR-HSI (visualized as RGB), and the bottom row shows the corresponding difference maps between the reconstructed results and the reference Ground Truth.

degradations such as noise, misalignment, and artifacts into HR-MSI leads to a more pronounced performance drop. This suggests that our method relies on the quality of HR-MSI, which is consistent with our structure tensor design based on the texture details of HR-MSI and therefore matches our experimental expectation. In addition, spectral variation in LR-HSI also causes a relatively large impact, indicating that our method depends less on the spatial quality of LR-HSI and more on the spectral quality of LR-HSI and the spatial quality of HR-MSI.

## I. Complexity Analysis

Firstly for convenience, we set $H$ and $W$ to denote the spatial dimensions of the high-resolution image, $C$ the number of spectral bands of the hyperspectral image, and $c$ the number of spectral bands of the multispectral image, where typically $c \ll C$. We set the $D$ as the hidden layer dimensions, $K$ as the number of steps and $k$ as the kernel size of convolutions (a little different from the main text). The time complexity of constructing the structure-induced Riemannian geometry involves gradient calculation and tensor smoothing on the MSI, which is $O(HWc)$. The time complexity of the Structure-Aware Flow Matching network is dominated by the feature encoding and fusion modules. Specifically, the spectral and auxiliary encoders involve convolutions with complexity $O(HW(CD + cD)k^2)$, and the lightweight fusion module with point-wise and depth-wise convolutions has a complexity of approximately $O(HWD^2)$. The calculation of the Riemannian Flow Matching loss and structure-aware optimal transport direction requires pixel-wise matrix operations and inversions over the spectral vectors, with a time complexity of $O(HWC)$. Therefore, the total time complexity of our SA-RFM training per iteration is $O\big(HW(CD + D^2)\big)$. For inference, given $K$ steps of the ODE solver, the total time complexity is $O\big(KHW(CD + D^2)\big)$.

The input of the framework includes the LR-HSI, HR-MSI, and the constructed Riemannian metric tensor $M \in \mathbb{R}^{H \times W \times 2 \times 2}$, and the space complexity for storing these inputs and the generated High-Resolution HSI is $O(HWC)$. The weight matrices of the network, primarily in the convolutional layers of the encoders and the fusion block, contain parameters proportional to $O(CD + D^2)$. The intermediate feature maps generated during the forward pass require memory space of $O(HWD)$. Therefore, the total space complexity of SA-RFM is $O\big(HW(C + D) + D^2\big)$, which demonstrates that the memory consumption grows linearly with the spatial resolution and channel dimension, ensuring scalability.

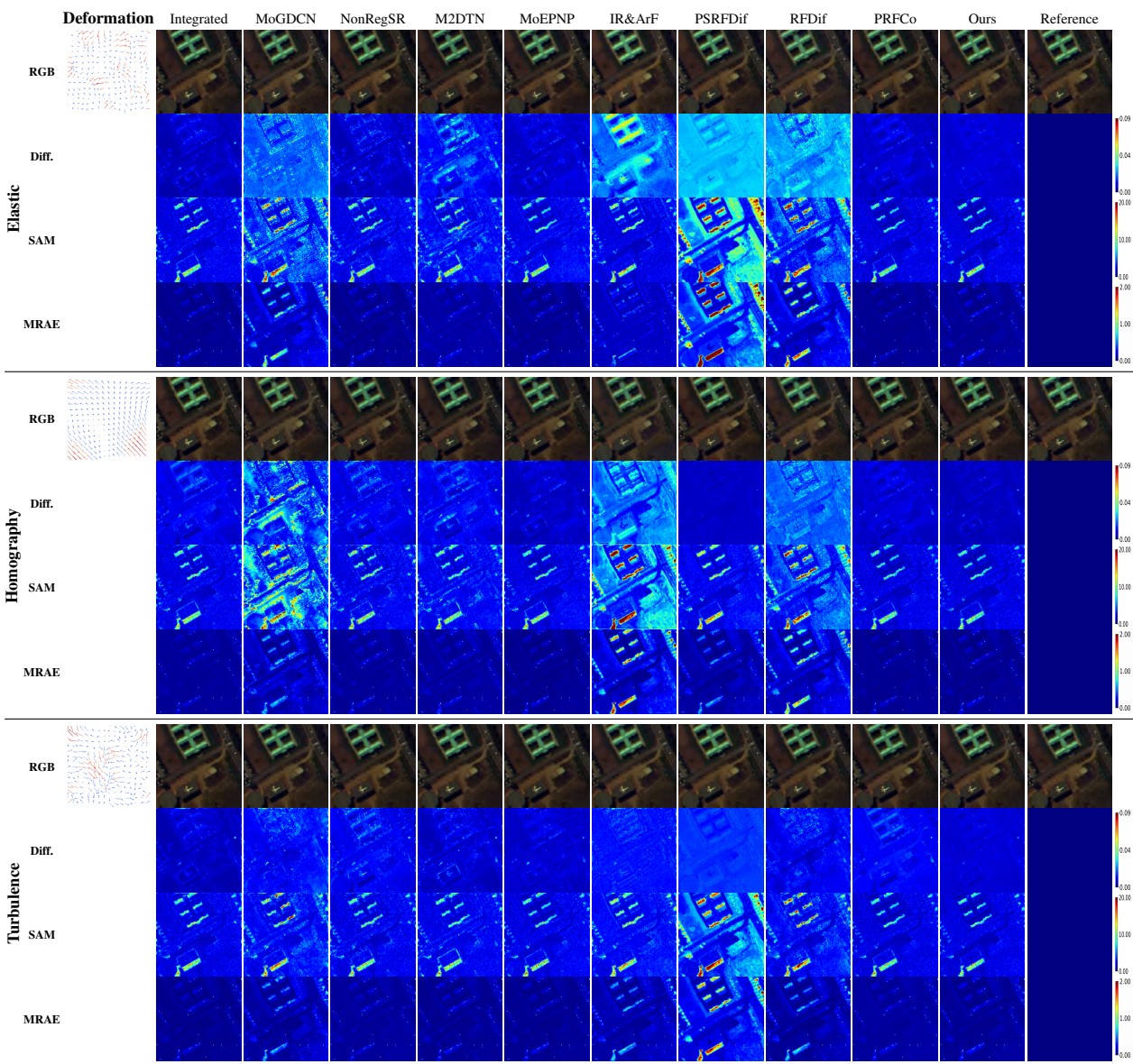

*Figure 7.* Visual performance comparison under three types of distortions on the Pavia University dataset. The first row shows estimated RGB (bands 67-29-1) images, the second row shows the difference maps averaged over all bands with respect to the reference, the third row shows the Spectral Angle Mapper (SAM) error maps, and the fourth row shows the Mean Relative Absolute Error (MRAE) maps, the first column displays the Deformation field. Far-right colorbars indicate the error ranges.

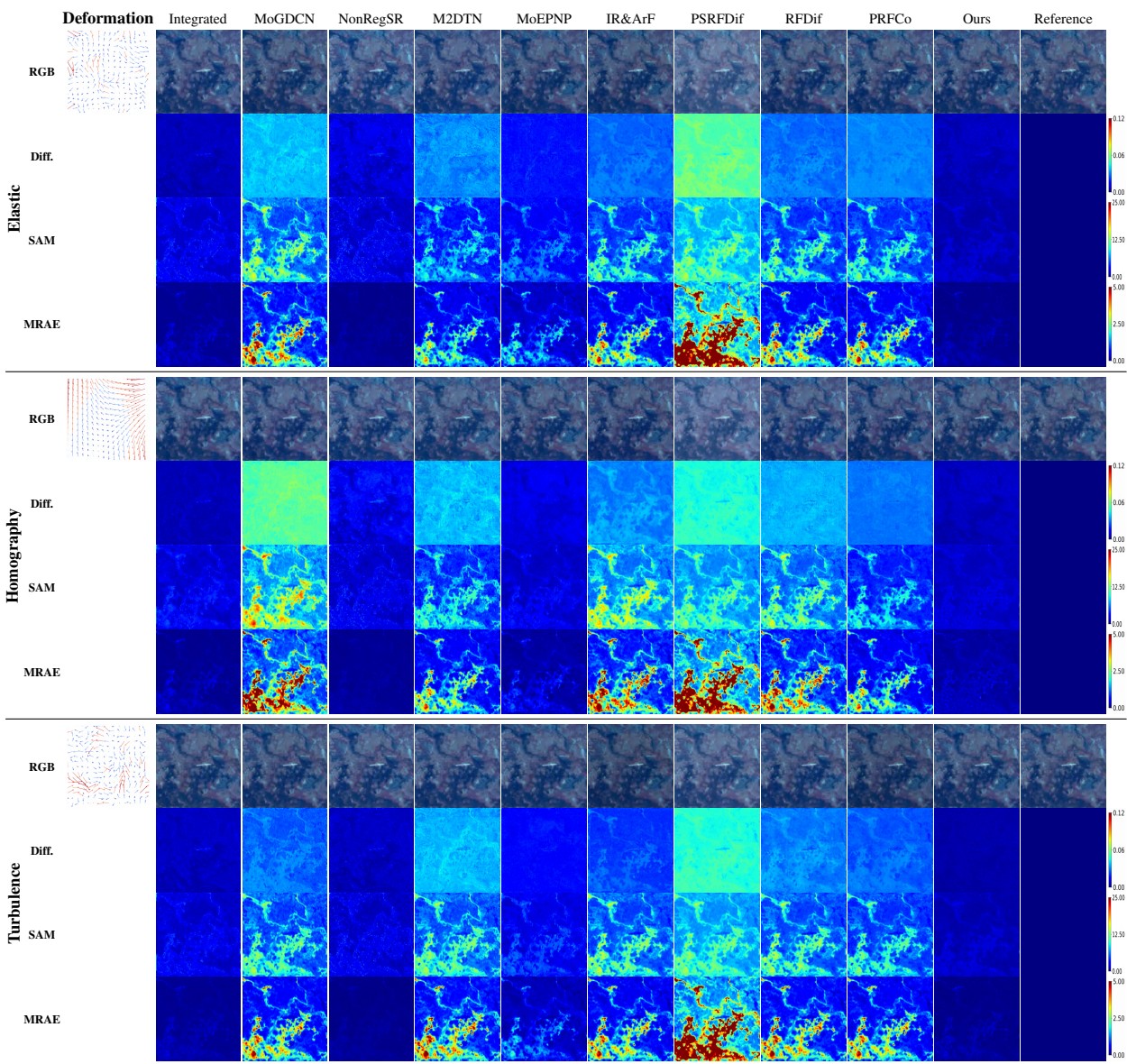

*Figure 8.* Visual performance comparison under three types of distortions on the Botswana dataset. The first row shows estimated RGB (bands 48-15-4) images, the second row shows the difference maps averaged over all bands with respect to the reference, the third row shows the Spectral Angle Mapper (SAM) error maps, and the fourth row shows the Mean Relative Absolute Error (MRAE) maps, the first column displays the Deformation field. Far-right colorbars indicate the error ranges.

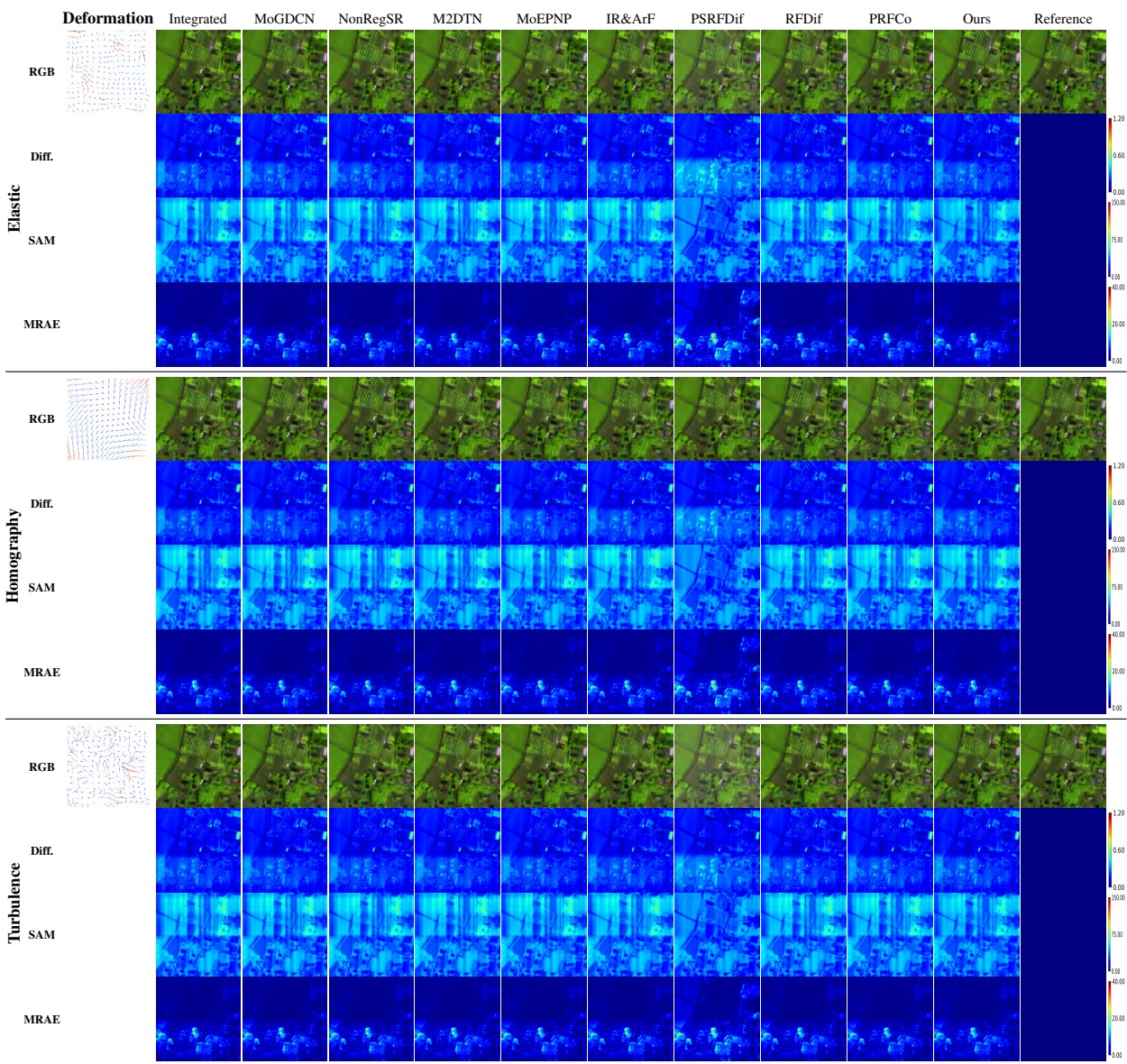

*Figure 9.* Visual performance comparison under three types of distortions on the Chikusei dataset. The first row shows estimated RGB (bands 101-71-37) images, the second row shows the difference maps averaged over all bands with respect to the reference, the third row shows the Spectral Angle Mapper (SAM) error maps, and the fourth row shows the Mean Relative Absolute Error (MRAE) maps, the first column displays the Deformation field. Far-right colorbars indicate the error ranges.

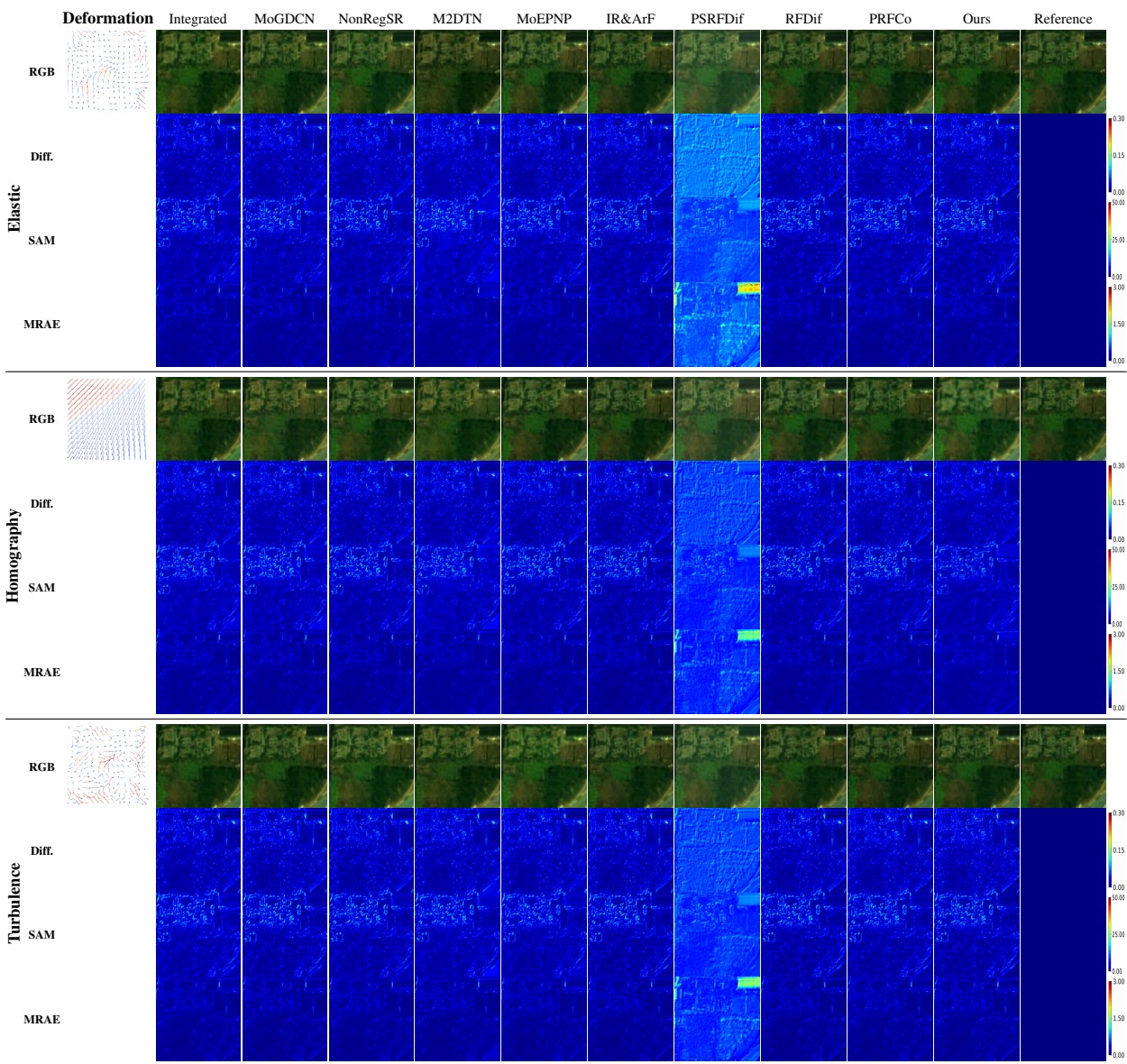

*Figure 10.* Visual performance comparison under three types of distortions on the Xiongan dataset. The first row shows estimated RGB (bands 103-77-52) images, the second row shows the difference maps averaged over all bands with respect to the reference, the third row shows the Spectral Angle Mapper (SAM) error maps, and the fourth row shows the Mean Relative Absolute Error (MRAE) maps, the first column displays the Deformation field. Far-right colorbars indicate the error ranges.

