# OpenReview forum: "Structure-Aware Riemannian Flow Matching for Registration and Fusion of Hyperspectral and Multispectral Images"
_ICML.cc/2026/Conference — ICML 2026 regular_

### Official Review · Reviewer_GsLL · 2026-02-17

**Soundness:** 3
**Presentation:** 4
**Significance:** 3
**Originality:** 4
**Overall Recommendation:** 5
**Confidence:** 5

**Summary:**

This is a well-structured paper with comprehensive experimental validation. The authors provide a clear formulation of the hyperspectral and multispectral image registration and fusion problem by interpreting it as a Riemannian Optimal Transport (OT) problem and propose a principled solution. Extensive comparisons and ablation studies convincingly demonstrate the effectiveness of the proposed method.

**Compliance With Llm Reviewing Policy:**

Affirmed.

**Final Justification:**

The authors solved my concerns, and I rise the rating to 5.

**Key Questions For Authors:**

See weakness.

I will rise my rating if the weaknesses are solved.

**Limitations:**

See weakness.

**Strengths And Weaknesses:**

## Strength

1.The authors reinterpret hyperspectral and multispectral image registration and fusion as a Riemannian OT problem and approximate it using flow matching. They further identify the mismatch between Euclidean supervision and anisotropic metrics, and introduce the $L_{OTD}$ to correct the transport direction. The introduction clearly presents the background and motivation, and the overall logical flow of the paper is coherent and well organized.
2.The authors conduct thorough ablation studies, including different constructions of the Riemannian metric, comparisons between Riemannian and Euclidean formulations, $L_{RFM}$ vs. $L_{OTD}$, sampling steps, and extensive parameter sensitivity analyses. The experimental results are comprehensive and convincingly validate the effectiveness of the proposed method.
3.The paper includes extensive comparisons with state-of-the-art registration and fusion methods, covering nearly all recent publicly available approaches, especially diffusion-based models. In addition to comparisons across multiple datasets, Table 3 provides a detailed comparison of efficiency and the number of sampling steps among diffusion-based methods, which is particularly informative.
4.Experiments are conducted on four datasets under three types of distortions, making the evaluation highly comprehensive. Overall, the proposed method outperforms all competing methods in most scenarios. In the few cases where it does not achieve the best performance, it remains highly competitive and achieves second-best results.

## Weakness

1.In Eq. (7), the authors introduce a hyperparameter to control the strength of the structure tensor components. Although extensive ablation and sensitivity analyses are provided in Table 4, Table 5, and Figure 4, it may be worth exploring whether this parameter could be learned automatically instead of being manually tuned.
2.Based on the authors’ formulation, the Riemannian projection could potentially be learned via a neural network and integrated into the backbone architecture. The authors are encouraged to provide a theoretical analysis showing whether $L_{OTD}$ is equivalent to a Riemannian projection, or to clarify the relationship between the two and discuss their respective advantages and limitations.

---

> ### Author Rebuttal · Authors · 2026-03-29
>
> We appreciate your positive comments regarding the clarity of our paper's structure, as well as the comprehensiveness of our experiments and evaluations. We provide the following responses to your questions.
>
> **1. Adaptive Learning of Hyperparameter $\beta$**
>
> We appreciate this very interesting suggestion. We have now supplemented our work with this experiment; specifically, we use a one-layer MLP for the direct prediction of hyperparameter $\beta$. The corresponding ablation result is summarized in Table 1 (2nd row) in anonymous link (https://anonymous.4open.science/r/submission-12123/1.pdf). In fact, the results are comparable to those obtained through hyperparameter tuning, yielding only a marginal improvement. This discussion will be included in the revised appendix as a reference for future readers.
>
> **2. Theoretical Relationship and Clarification**
>
> Mathematically, our $v^{\dagger}(u)$ in Eq. (14) is the analytical Riemannian projection of the Euclidean displacement $d(u)$ onto the tangent space of the structure-induced manifold, subject to the inner product constraint. Consequently, $L_{OTD}$ is not a substitute for the projection, but rather the learning mechanism used to supervise the neural network $v_{\theta}$ to approximate this Riemannian optimal direction. Instead of performing a hard projection of the network's output during every forward pass, we use $L_{OTD}$ to guide the velocity field toward the manifold's geometry during training.
>
> * **Pros of our $L_{OTD}$ approach:**
> Acting as a "soft constraint," $L_{OTD}$ provides a smoother loss landscape. Hard projections can lead to optimization instability or vanishing gradients if the metric $M(u)$ is locally ill-conditioned due to MSI noise. The regularization weight $\lambda_{OTD}$ allows the model to flexibly balance global Euclidean consistency with local structural optimality, which is crucial for handling complex non-rigid deformations. By training the network to internalize the Riemannian geometry, we maintain high inference speed without requiring explicit matrix inversions or projections during the sampling process.
>
> * **Cons of integrated Riemannian Projection:**
> Explicitly integrating a projection layer into the backbone would require repeated matrix operations ($M^{-1}$) at every pixel in every training iteration, increasing the computational burden.  A hard projection forces the flow to strictly follow MSI structural cues, making the system highly sensitive to any registration errors or noise present in the MSI. It is worth noting that we deliberately apply Gaussian blurring to the MSI textures in Eq. (6) to mitigate excessive sensitivity to minor textural variations, thereby enhancing the model's robustness.
>
> In summary, while $v^{\dagger}$ is the Riemannian projection, supervising the network via $L_{OTD}$ is numerically superior for deep learning as it ensures robust convergence and maintains computational efficiency.

---

> > ### Author Rebuttal · Reviewer_GsLL · 2026-04-01
> >
> > The rebuttal solved my concerns. I will rise the score to 5.

---

> > > ### Author Response · Authors · 2026-04-07
> > >
> > > Thank you very much for your professional feedback and suggestions for improving our manuscript. We will supplement the relevant experiments in the appendix and add the analysis.

---

### Official Review · Reviewer_ptbb · 2026-03-06

**Soundness:** 3
**Presentation:** 3
**Significance:** 3
**Originality:** 3
**Overall Recommendation:** 5
**Confidence:** 3

**Summary:**

This paper proposes a novel framework, SA-RFM, for unregistered hyperspectral and multispectral image registration and fusion. The authors formulate the registration-fusion problem as a transport process guided by a Riemannian metric derived from the structural tensor of the high-resolution MSI. The framework integrates two key loss functions: RFM and OTD regularization, enabling structure-aware deformation learning. Extensive experiments on multiple datasets under various distortions demonstrate the effectiveness and efficiency of the proposed method, achieving state-of-the-art performance with significantly lower computational cost.

**Compliance With Llm Reviewing Policy:**

Affirmed.

**Final Justification:**

All my concerns have been addressed in the rebuttal.

**Key Questions For Authors:**

1.	In Eq. (6), the structure tensor is computed from the MSI using Sobel gradients and Gaussian smoothing. How sensitive is the performance to the choice of gradient operator and smoothing parameters? Please provide more insight into the robustness of this construction.
2.	The OTD regularization in Eq. (13) imposes a directional constraint on the velocity field. How does this interact with the Riemannian metric in practice? Could there be cases where the optimal direction conflicts with the structure-aware constraint?
3.	The method assumes that the HR-MSI provides reliable structural priors. What happens when the MSI itself contains noise, artifacts, or misalignments? How does the model cope with such imperfections?

**Limitations:**

There is no explicit discussion about limitations. The following discussion would further improve the manuscript.
1. The current framework relies on the availability of a high-resolution MSI with clear structural information. In scenarios where the MSI is of low quality or has different spectral sensitivity, the constructed Riemannian metric may not be reliable.
2. While the one-step inference is efficient, the training process involves constructing the Riemannian metric and computing matrix inversions, which may introduce computational overhead in the training phase.

**Strengths And Weaknesses:**

Strengths
1. The paper introduces a geometry-aware perspective to the HSI-MSI fusion problem by embedding Riemannian metric learning into a flow matching framework.
2. The method is evaluated on multiple datasets with different spectral and spatial resolutions, under three types of geometric distortions. The comparisons with state-of-the-art methods are thorough and well-documented.
3. The authors provide extensive ablation studies on loss functions, metric construction, backbone networks, and sampling steps, offering valuable insights into the contribution of each component.

Weaknesses
1. Some mathematical notations are introduced without sufficient explanation (e.g., the push-forward operator T_# in Eq. (4), and the conditional flow matching formulation).
2. Limited Discussion on Failure Cases: While the method performs well overall, the paper does not discuss scenarios where it might fail (e.g., extreme occlusions, low-texture regions, or severe spectral variability).
3. The method is tailored for HSI-MSI fusion. It remains unclear whether the framework can generalize to other multi-modal registration tasks (e.g., RGB-depth, MRI-PET). A brief discussion on this would be beneficial.
4. Several typos in the manuscript, please check (e.g., P4, line 203).

---

> ### Author Rebuttal · Authors · 2026-03-29
>
> We appreciate your recognition of the adequacy of our comparative experiments and the comprehensiveness of our ablation studies.
>
> **1. Clarification of Notations and Correction of Typos**
>
> We thank you for the careful correction. We have updated the manuscript to provide explicit definitions for all notations. Specifically, T_\# denotes the push-forward operator, which describes the transformation of a probability measure via a measurable map $T$. Additionally, we have corrected the typo on Line 203 and conducted a comprehensive, line-by-line review of the entire paper to rectify similar errors and ensure linguistic precision.
>
> **2. Robustness to MSI Imperfections and Analysis of Failure Cases**
> In fact, the $G_{k,\sigma}$ term in Eq. (6) possesses a certain degree of anti-interference capability. $G_{k,\sigma}$ is applied to the MSI textures; the original intention of introducing this term was to reduce sensitivity to fine texture details in the MSI, allowing the model to focus on more macroscopic structural features and preventing trivial details from affecting convergence. We designed a sensitivity analysis for the blur kernel as shown in Fig. 5, which can be viewed as an evaluation of robustness to MSI with different blurring levels (analogous to texture changes caused by noise, artifacts, etc., in the MSI). In principle, our method is less applicable in scenarios involving extreme occlusions or overly smooth MSI textures. However, from a theoretical perspective, current registration frameworks generally exhibit poor adaptability to such extreme cases.
>
> We have supplemented our study with sensitivity experiments on MSI, covering noise, artifacts, and misalignment, as detailed in Figure 1 and Table 1 of the anonymous link (https://anonymous.4open.science/r/submission-12123/1.pdf). The results indicate that our method exhibits strong robustness to noise and moderate performance on artifacts, while showing relatively limited efficacy in handling misalignment.
>
> **3. Generalizability to other multi-modal tasks**
>  Theoretically, our framework possesses strong generalizability and can be extended to other multi-modal registration tasks. The key to adaptation lies in clear role assignment:
>
> * **MSI Role**: Choose the modality with high spatial resolution and rich structural details (e.g., RGB in RGB-D, or MRI in MRI-PET) to construct the structure-induced Riemannian metric  that guides the flow field.
> * **HSI Role**: The modality that requires geometric alignment and deformation.
>
> Additionally, the framework’s flexibility allows for customizing the cost function to incorporate task-specific physical properties or geometric priors. For example, in medical imaging, one could enforce local volume preservation, while in RGB-D tasks, depth continuity could be prioritized. Tailoring the cost function to these specific task characteristics will yield superior results. We will add a brief discussion on this generalizability in the revised manuscript.
>
> **4. Gradient Operators and Smoothing Parameters in Eq. (6)**
>
> In fact, we have already conducted these experiments in the original manuscript. Specifically, Table 4 presents a sensitivity analysis regarding the choice of gradient operators (Scharr and Prewitt) , while Figure 5 illustrates the impact of parameter selection for the Gaussian smoothing kernel on performance.
>
> **5. Interaction between OTD Regularization and the Riemannian Metric**
>
> We thank you for this insightful question. The OTD direction $v^\dagger$ is the analytical solution to Eq. (13), serving as a geometric compromise between global Euclidean progress and local structural constraints:
>
> $v^\dagger$ is mathematically defined as the unique vector that minimizes the Riemannian cost $v^\top M(u) v$ while strictly maintaining the required displacement toward the target. Instead of a "conflict," $v^\dagger$ finds the path of least resistance, warping the trajectory to follow structural low-cost paths (e.g., along edges) to avoid crossing high-cost boundaries.
>
>  In practice, the regularization weight $\lambda_{OTD}$ ensures a stable interaction. This prevents the model from over-fitting to local structures at the expense of global registration accuracy.
>
> In summary, $v^\dagger$ acts as a geometrically-informed guide that internalizes the Riemannian manifold into the velocity field without sacrificing convergence.
>
> **6. Training Efficiency and Computational Overhead**
>
> The training overhead of our method is marginal. As detailed in the Complexity Analysis (Appendix G), our Riemannian operations involve only closed-form $2 \times 2$ matrix solutions (e.g., analytical inversion and eigendecomposition) at each pixel. This preserves the complexity of standard Flow Matching.

---

> > ### Author Rebuttal · Reviewer_ptbb · 2026-04-02
> >
> > The rebuttal solved my concerns. I will raise my score to 5.

---

> > > ### Author Response · Authors · 2026-04-07
> > >
> > > Thank you very much for your suggestions. Your feedback is very important for us to improve the manuscript. We will include the supplementary experiments in the appendix of the final version. We will also correct the Notations in the manuscript.

---

### Official Review · Reviewer_qnfr · 2026-03-10

**Soundness:** 2
**Presentation:** 2
**Significance:** 3
**Originality:** 3
**Overall Recommendation:** 4
**Confidence:** 4

**Summary:**

Overall, the authors discuss the issue of joint registration and fusion of hyperspectral and multispectral images, a critical prerequisite for high-resolution hyperspectral imaging. The paper proposes Structure-Aware Riemannian Flow Matching (SA-RFM), which reformulates registration as dynamic optimal transport on a structure-induced Riemannian manifold. The key innovation is incorporating anisotropic transport costs derived from MSI structural cues into a conditional flow matching framework, avoiding the computational intractability of explicit OT solvers. Two complementary mechanisms are introduced: (1) a Riemannian Flow Matching objective for structure-aware error measurement, and (2) an optimal transport direction regularization that aligns the velocity field with the induced metric.

**Compliance With Llm Reviewing Policy:**

Affirmed.

**Key Questions For Authors:**

1. How sensitive is the method to the quality of MSI structural cues? What happens when the MSI has poor texture or noise?
2. What is the computational complexity compared to explicit OT solvers and standard flow matching?
3. How does the local approximation in Eq. (5) affect accuracy for large deformations?
4. Can you provide complete quantitative results with standard fusion metrics?

**Limitations:**

yes

**Strengths And Weaknesses:**

1. Soundness

Strengths:
- The theoretical foundation linking optimal transport, Riemannian geometry, and flow matching is sound
- The structure tensor-based metric construction (Eq. 2-3) is well-established and appropriate
- The local cost approximation (Eq. 5) is a clever way to avoid expensive geodesic computations

Weaknesses:
- The Riemannian metric tensor G(u) is constructed from MSI structure, but the sensitivity to MSI quality is not discussed
- The approximation in Eq. (5) assumes local linearity—how does this affect registration accuracy for large deformations?
- The computational complexity analysis is missing; how does SA-RFM compare to explicit OT solvers?
- No analysis of failure cases or limitations

2. Presentation

Strengths:
- The conceptual comparison in Figure 1 effectively illustrates the approach
- The mathematical notation is clear and consistent
- The motivation for geometry-aware costs is well-articulated

Weaknesses:
- Some mathematical details (e.g., the exact form of the structure tensor) could be more explicit
- The connection between the Riemannian metric and the flow matching objective could be explained more intuitively

3. Originality and Significance

Strengths:
- The formulation of image registration as dynamic optimal transport on a structure-induced Riemannian manifold is novel and theoretically grounded
- The identification of the "fundamental mismatch between anisotropic transport costs and conventional Euclidean displacement supervision" is a key insight
- The local cost approximation to avoid explicit OT solvers is a practical and elegant solution
- The integration of flow matching with geometry-aware costs represents a meaningful advancement over standard diffusion-based registration

Weaknesses:
- Flow matching for image registration has been explored in concurrent work; the novelty lies primarily in the Riemannian geometry aspect
- The anisotropic cost construction from MSI structural cues, while effective, follows established techniques in structure tensor analysis

---

> ### Author Rebuttal · Authors · 2026-03-29
>
> We thank the reviewer for recognizing our motivations, the theoretical framework, and the clarity of our figures and formulas. We answer your questions as follows.
>
> **1.Sensitivity on MSI quality**
>
> As shown in Eq. (6), a Gaussian blur kernel is integrated to enhance robustness against blurred MSI boundaries. Figure 5 further validates this design: overly precise textures can cause optimization oscillations, while excessive blurring results in a performance drop. See Figure 1 and Table 1 in anonymous link (https://anonymous.4open.science/r/submission-12123/1.pdf). It can be observed that our method is less effective in handling MSI misalignment, whereas it remains relatively insensitive to noise or artifacts.
>
> **2.Sensitivity on large distortion on LR-HSI**
>
> As shown in the 4th response to Reviewer HhPU, performance degradation caused by large HSI deformations can be mitigated by increasing the number of inference steps. See the 4th row in Table 1 in anonymous link (https://anonymous.4open.science/r/submission-12123/1.pdf). It can be observed that our method is insensitive to stronger elastic distortion on LR-HSI, even in the one-step scenario.
>
> **3.Computational complexity compared to explicit OT solvers and standard flow matching**
>
> As detailed in Appendix Sec. G, the computational complexity of SA-RFM is practically identical to standard flow matching. Our training time complexity per iteration is $\mathcal{O}(HW(CD+D^2))$. The only additional operation is constructing the Riemannian metric from the MSI, which introduces a minimal and lightweight overhead of $\mathcal{O}(HWc)$ (where $c \ll C$).
>
> In contrast, explicit OT solvers or exact geodesic computations are computationally intractable for this task. They require solving complex Boundary Value Problems (BVPs) or performing expensive Sinkhorn iterations across millions of pixels at every single training step. Our local cost approximation effectively bypasses this massive computational bottleneck while successfully maintaining geometry-aware transport.
>
> **4. Sensitivity and Limitations**
>
> Since the design of our structural tensor relies on the MSI, our model exhibits greater robustness to variations in the HSI, while being relatively more sensitive to changes in the MSI. We have provided additional experiments conducted under conditions of poor HSI and MSI quality, as shown in Figure 1 and Table 1 of the anonymous link (https://anonymous.4open.science/r/submission-12123/1.pdf). Furthermore, because we assume in Eq. (7) of the manuscript that distortion can be decomposed into a weighted combination of isotropic and anisotropic components, this introduces a certain degree of error (as real-world distortions often involve complex, non-linear interactions). Although our experimental results verify that the proposed method consistently maintains state-of-the-art performance, future researchers could propose more robust improvements addressing these two specific points to further advance our methodology.
>
> **5.Exact Formulation of the Structure Tensor**
>
> We compute the MSI grayscale gradients $\nabla I=[I_x,I_y]^\top$ and the base tensor $J_0=\nabla I\nabla I^\top$. We extract its principal eigenvalue $\lambda_1$ and unit eigenvector $e_1=[e_x,e_y]^\top$, with orthogonal vector $e_2=[-e_y,e_x]^\top$.
>
> To penalize transport across strong edges, we non-linearly amplify the principal eigenvalue $\lambda_1'=1+\alpha\lambda_1^k$ (using $\alpha=10,k=2$), while keeping $\lambda_2'=1$. The exact metric tensor in our code is:
>
> $$M(u)=\lambda_1'e_1e_1^\top+e_2e_2^\top=\begin{bmatrix}\lambda_1'e_x^2+e_y^2&(\lambda_1'-1)e_xe_y/(\lambda_1'-1)e_xe_y&\lambda_1'e_y^2+e_x^2\end{bmatrix}$$
>
> We will update our definition in the revised manuscript.
>
> **6. Originality regarding Flow Matching and Structure Tensors**
>
> Thank you for your concerns. Our core novelty is not inventing Flow Matching or structure tensors, but elegantly synthesizing them to solve multi-modal HSI-MSI registration.  Existing flow matching relies on isotropic Euclidean metrics, which ignore image topology and cause spectral mixing across boundaries. Meanwhile, classical structure tensors are absent from continuous ODE generative models.  Our primary innovation lies in embedding the structure tensor to redefine dynamic Optimal Transport as a Riemannian manifold. Crucially, our proposed Optimal Transport Direction (OTD) regularization resolves the fundamental mismatch between anisotropic transport costs and conventional Euclidean supervision. This specific integration provides a geometry-aware mechanism lacking in standard frameworks.
>
> **7. Experiment on registered HSI and MSI fusion (only fusion)**
>
> We have provided the fusion results for distortion-free LR-HSI and HR-MSI, as shown in the 3rd row of Table 1 in the anonymous link (https://anonymous.4open.science/r/submission-12123/1.pdf). The results demonstrate that our method outperforms the baselines, which is consistent with our expectations.

---

> > ### Author Rebuttal · Reviewer_qnfr · 2026-04-01
> >
> > Thanks for the author's reply. I hope the author can make these significant modifications in the future version, and I remain my original score.

---

> > > ### Author Response · Authors · 2026-04-07
> > >
> > > Thank you very much for your professional feedback. Your comments are crucial for the revision of our manuscript, and we will update the above content in the future version.

---

### Official Review · Reviewer_HhPU · 2026-03-12

**Soundness:** 3
**Presentation:** 3
**Significance:** 3
**Originality:** 3
**Overall Recommendation:** 5
**Confidence:** 4

**Summary:**

In this paper, the authors tackle the problem of multispectral and hyperspectral image fusion. The proposed solution is a structure-aware Riemannian flow matching procedure. To achieve this, they compute a Riemannian metric that captures the structural boundaries or directions that will guide the flow in an anisotropic manner. This Riemannian metric is incorporated into the Flow Matching loss, resulting in a structure-aware objective. Furthermore, they combine this Flow Matching loss with an Optimal Transport regularization to force the velocity field to also respect the structure of the captured terrain. To evaluate the proposed method, extensive experiments were conducted on multiple datasets, with comparisons to a large number of state-of-the-art methods. A substantial number of ablation studies are also included to demonstrate the importance of each component of the proposed method.

**Compliance With Llm Reviewing Policy:**

Affirmed.

**Final Justification:**

This paper introduces an interesting method for fusing multispectral and hyperspectral images using a Riemannian metric. The experimental analysis is strong, and the authors' rebuttal has helped to clarify several theoretical points. Therefore, I recommend this paper for acceptance at ICML 2026.

**Key Questions For Authors:**

1. The whole point of the paper is to introduce a Riemannian metric that captures the local structure in the hyperspectral images. You add this Riemannian metric in the loss of the Flow Matching but you still rely on Euclidean displacement. In order to counter this, you add an optimal transport direction regularization. Why did you not consider a fully Riemannian Flow Matching framework like the one proposed in [1]? In the paper of Lipman et al. they propose to use geodesics to compute the displacement, leading to a fully Riemannian setup. I note your claim in Section 1 that computing geodesics on high-dimensional image manifolds is computationally intractable. Could you elaborate on this specific bottleneck given your chosen metric? Could a fully Riemannian setup theoretically eliminate the need for the heuristic OTD regularization, and if so, is there a middle ground for approximating those geodesics?
2. You approximate the Riemannian energy by freezing the metric at $u$ and assuming a straight-line path. While you provide an empirical ablation for this (Table 5), do you have any theoretical discussion regarding the approximation error or the bounds of this simplification, especially for large deformations?


[1] Lipman, Y., Chen, R. T., Ben-Hamu, H., Nickel, M., and Le, M. Flow matching for generative modeling. arXiv preprint arXiv:2210.02747, 2022.

**Limitations:**

Yes

**Strengths And Weaknesses:**

# Soundness

Overall, the paper is pretty technically sound, both on the theoretical side and the experimental side. Let me detail each side:
- On the theoretical side, one could expect nevertheless more rigour in the analysis:
   - In equation (9), a $\rho_t(u)$ appears to be missing in the definition of $\mathcal{J}(v_t)$. Including this term is necessary to ensure consistency with the standard Benamou-Brenier formula provided in Equation (2). Furthermore, the manuscript lacks a clear theoretical derivation or justification for this specific formulation of structure-aware kinetic energy.
   - In section 3.3, the authors claim that the problem given at Equation (13) admits a closed-form solution that they give at equation (14). One could expect the detail of this proof in the appendix.

- The experimental analysis is highly rigorous. The authors evaluate their approach against a comprehensive set of state-of-the-art baselines across multiple distortion types. Additionally, all newly introduced components and hyperparameters are deeply investigated through extensive ablation studies.

# Presentation

The paper is well written and easy to read. Even if the _Related works_ section is in the appendix, this paper clearly positions itself in the existing literature.


# Significance
 Introducing anisotropy in Flow Matching velocities is a meaningful contribution, as it enables directional behavior that better aligns with image structures. The resulting improvements suggest that the method addresses a genuine limitation of the standard flow matching methods. However, the method is evaluated on a very specific application, and it would be interesting to see how the proposed approach would benefit other tasks.

# Originality
 The use of structural information to define anisotropic metrics is well established in image processing (e.g., through the structure tensor). Integrating such a metric within the Flow Matching framework is novel.

---

> ### Author Rebuttal · Authors · 2026-03-29
>
> We thank you for your positive assessment, particularly regarding the technical soundness, experimental validation, and originality of our work. We address your questions below:
>
> **1. Missing $\rho_t(u)$ in Equation (9) and theoretical derivation**
>
> We have corrected Eq. (9) to include $\rho_t(u)$ for consistency with the Benamou-Brenier formula:
>
> $$\mathcal{J}(v_t)=\int_{0}^{1}\int_{\Omega}v_t(u)^\top M(u)v_t(u)\rho_t(u)dudt$$
>
> We will add its theoretical derivation to the revised appendix as follows.
>
> The standard dynamic Optimal Transport minimizes Euclidean kinetic energy:
>
> $$\inf_{v_t}\int_0^1\int_\Omega\frac{1}{2}\Vert v_t(u)\Vert_2^2\rho_t(u)dudt$$
>
> To penalize unnatural transport across structural boundaries, we redefine the spatial domain $\Omega$ as a Riemannian manifold with our position-dependent metric tensor $M(u)=I+\beta S_Y(u)$. The local kinetic energy becomes the squared Riemannian norm:
>
> $$\Vert v_t(u)\Vert_{M(u)}^2=v_t(u)^\top M(u)v_t(u)$$
>
> Substituting this into the optimal transport objective yields our structure-aware kinetic energy $\mathcal{J}(v_t)$. This mathematically forces the transport trajectory to align with image structures, as crossing boundaries (where $M(u)$ has large eigenvalues) incurs a severe kinetic penalty.
>
> **2. Proof of the closed-form solution for Eq. (13) and Eq. (14)**
>
> We will include this in the appendix. Equation (13) is a quadratic program with a linear constraint: $\min_{v}v^\top M(u)v$ subject to $d(u)^\top v=\Vert d(u)\Vert_2$.
>
> We construct the Lagrangian $\mathcal{L}(v,\lambda)=v^\top M(u)v+\lambda(\Vert d(u)\Vert_2-d(u)^\top v)$. Setting its gradient to zero ($\nabla_v\mathcal{L}=2M(u)v-\lambda d(u)=0$) and utilizing the invertibility of $M(u)$ yields:
>
> $$v=\frac{\lambda}{2}M(u)^{-1}d(u)$$
>
> Substituting this $v$ into the original constraint $d(u)^\top v=\Vert d(u)\Vert_2$ allows us to solve for the scalar $\frac{\lambda}{2}$:
>
> $$\frac{\lambda}{2}\left(d(u)^\top M(u)^{-1}d(u)\right)=\Vert d(u)\Vert_2\implies\frac{\lambda}{2}=\frac{\Vert d(u)\Vert_2}{d(u)^\top M(u)^{-1}d(u)}$$
>
> Plugging $\frac{\lambda}{2}$ back into our expression for $v$ yields the exact closed-form solution in Eq. (14):
>
> $$v^\dagger(u)=\frac{M(u)^{-1}d(u)}{d(u)^\top M(u)^{-1}d(u)}\Vert d(u)\Vert_2$$
>
> **3. Fully Riemannian Flow Matching vs. OTD Regularization**
>
> Thank you for this profound theoretical inquiry. While a fully Riemannian Flow Matching (RFM) framework following exact geodesics is mathematically elegant, its application to high-dimensional image registration faces a severe computational bottleneck:
>
> Calculating exact geodesics requires solving the second-order ODE $\ddot{\gamma}^k+\Gamma^k_{ij}\dot{\gamma}^i\dot{\gamma}^j=0$, which depends on Christoffel symbols $\Gamma^k_{ij}$ derived from the spatial derivatives of the metric $M(u)$. Since our structure-induced metric is highly data-dependent and derived from discrete high-resolution gradients, solving this global Boundary Value Problem (BVP) across millions of pixels at every training iteration is computationally intractable.
>
> Our OTD regularization is explicitly designed as a computationally efficient surrogate for these geodesics. Instead of an expensive global BVP, OTD performs a local, first-order geometric correction. By analytically solving Eq. (13), the resulting $v^\dagger(u)$ acts as a tangent approximation: it maintains the global Euclidean direction to ensure convergence, while its local trajectory is warped by $M(u)^{-1}$ to favor minimum-cost structural paths.
>
> We will clarify this in the appendix.
>
> **4. Theoretical bounds and approximation error for the simplified Riemannian energy**
>
> By freezing the metric at $u$ and assuming a straight-line path for displacement $\Delta u$, we perform a first-order Taylor approximation of the Riemannian energy. The approximation error is bounded by:
>
> $$\text{Error}\leq\mathcal{O}(\Vert\nabla M(u)\Vert\Vert\Delta u\Vert^3)$$
>
> Since $M(u)=I+\beta S_Y(u)$, this bound is directly constrained by the smoothness (Hessian) of the multispectral image structure tensor $S_Y(u)$.
>
> You correctly notes that a straight-line assumption for large, single-step displacements causes the approximation error to blow up. However, our approach avoids this by operating within a continuous dynamic Flow Matching (ODE) paradigm. Rather than a single static leap, the total deformation is governed by $\frac{ds_t}{dt}=v_t(s_t)$. During inference, the ODE solver breaks the global deformation into discrete, infinitesimal integration steps $dt$, where the instantaneous displacement is exactly $\Delta u=v_tdt$. Since a small $dt$ keeps local displacement $\Vert\Delta u\Vert$ minimal, our quadratic approximation becomes asymptotically exact as $dt\to0$. Thus, the integration process prevents prohibitive error accumulation even under extreme macroscopic deformations. Furthermore, performance can be improved by increasing the number of steps, rather than being restricted to 1-step.

---

> > ### Author Rebuttal · Reviewer_HhPU · 2026-04-01
> >
> > I thank the authors for their detailed response. It addresses my various concerns, and including these clarifications in the appendix will improve the paper's soundness. I maintain my recommendation of 5 (Accept).

---

> > > ### Author Response · Authors · 2026-04-07
> > >
> > > Thank you very much for your recognition of our work and your professional suggestions. We will correct the revised content in the appendix.

---

### Decision · Program_Chairs · 2026-04-30

**Decision:**

Accept (regular)

**Comment:**

The paper addresses joint registration and fusion of hyperspectral and multispectral images by formulating registration as dynamic optimal transport on a structure-induced Riemannian manifold. A position-dependent metric derived from the MSI structure tensor defines anisotropic transport costs, which are integrated into a conditional flow matching framework through a Riemannian flow matching loss and an optimal transport direction regularization. The method is lightweight (1M params) and supports one-step inference.

All four reviewers (5, 4, 5, 5) marked their concerns fully resolved after rebuttal, with two raising scores. The method is well-validated across four datasets, three distortion types, and extensive ablations, achieving state-of-the-art results at low computational cost.